# VMDiff: Visual Mixing Diffusion for Limitless Cross-Object Synthesis

**Zeren Xiong**[1], **Yue Yu**[1], **Zedong Zhang**[1], **Shuo Chen**[2], **Jian Yang**[1,2] **& Jun Li**[1,2*]
[1]School of Computer Science and Engineering, Nanjing University of Science and Technology, China
[2]School of Intelligence Science and Technology, Nanjing University, China
{xiong_zeren, zandyz, yuue}@njust.edu.cn;
{shuo.chen, csjyang, istjunli}@nju.edu.cn;  * corresponding author

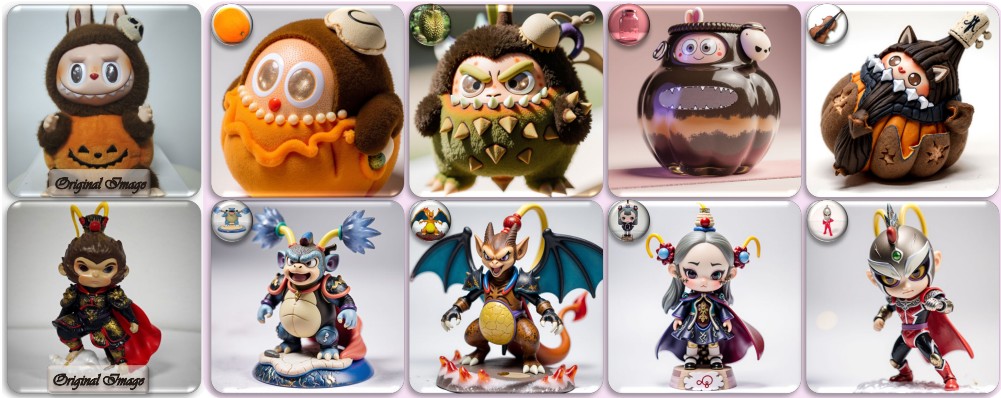

Figure 1: Two groups (rows) illustrating our VMDiff's capability to generate coherent hybrid objects. For each group, images from the 2*nd* to the 5*th* column are the product of fusing the source image in the 1*st* column with the corresponding image in the top left [2].

## Abstract

Creating novel images by fusing visual cues from multiple sources is a fundamental yet underexplored problem in image-to-image generation, with broad applications in artistic creation, virtual reality and visual media. Existing methods often face two key challenges: *coexistent generation*, where multiple objects are simply juxtaposed without true integration, and *bias generation*, where one object dominates the output due to semantic imbalance. To address these issues, we propose **Visual Mixing Diffusion (VMDiff)**, a simple yet effective diffusion-based framework that synthesizes a single, coherent object by integrating two input images at both noise and latent levels. Our approach comprises: (1) a ***hybrid sampling process*** that combines guided denoising, inversion, and spherical interpolation with adjustable parameters to achieve structure-aware fusion, mitigating coexistent generation; and (2) an ***efficient adaptive adjustment*** module, which introduces a novel similarity-based score to automatically and adaptively search for optimal parameters, countering semantic bias. Experiments on a curated benchmark of 780 concept pairs demonstrate that our method outperforms strong baselines in visual quality, semantic consistency, and human-rated creativity. Project.

## 1 Introduction

Synthesizing novel images by combining visual elements from multiple sources is a fundamental challenge in image-to-image generation, with wide applications in virtual reality (Haque et al., 2023;

---

[2]Our method's results, presented under the title *Creative Toys Series*, were awarded the Silver Award at the NY Digital Awards (Generative AI) 2025 (Award Page, Video).

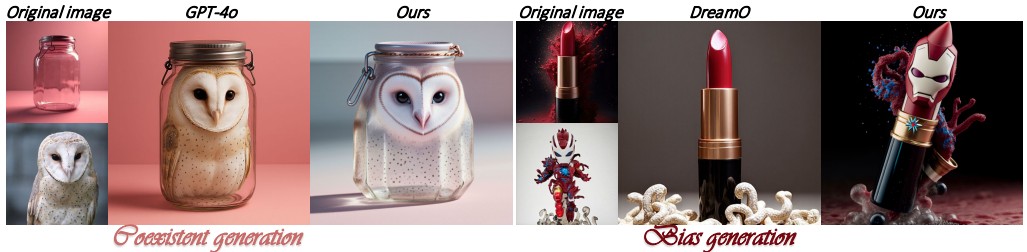

Figure 2: **Failed fusions between two object images.** GPT-4o OpenAI (2025) performs *coexistent generations* (left), while DreamO (Mou et al., 2025) exhibits *bias generations* (right). In contrast, our method achieves a seamless and harmonious fusion of the two objects.

Chen et al., 2024), digital media (Zheng et al., 2024; Zhao et al., 2024), product design (Ju et al., 2024; Sheynin et al., 2024; Wang et al., 2024) and film and game (Ceylan et al., 2023; Liu et al., 2024). In particular, visual composition methods generate high-fidelity images by composing objects through various strategies, such as combining object words into complex sentences (Liu et al., 2022), merging multiple objects (Liu et al., 2021), or blending scenes and styles (Zou et al., 2025). Although these approaches effectively position different objects or parts within an image, they often struggle to seamlessly integrate distinct elements into a single object. Recent semantic mixing (Li et al., 2024; Xiong et al., 2024) explores novel object synthesis by combining textual descriptions of one object with another images or text. In contrast, this work focuses on visual mixing—directly blending two object images into a single, imaginative, and visually cohesive concept.

However, when existing powerful methods are used to perform this visual mixing task, we identify two key limitations. First, **coexistent generation** (see Fig. 2, left) occurs when different objects merely appear in the same scene—either side-by-side or partially overlapped—without achieving true visual and semantic integration. While the resulting compositions are spatially coherent, they remain conceptually disjoint. For example, OpenAI's recent GPT-4o (OpenAI, 2025) produces an image where the glass jar and owl overlap but fail to meaningfully fuse. Second, **bias generation** (see Fig. 2, right) arises when the model generates only one object while omitting the other. This asymmetry likely stems from imbalanced representations or unresolved semantic conflicts, leading to outputs that disproportionately emphasize one object. For instance, DreamO (Mou et al., 2025) generates the lipstick while entirely neglecting the iron man figurine.

To address these limitations, we develop **Visual Mixing Diffusion (VMDiff)**, a simple yet effective framework for synthesizing novel, coherent objects that seamlessly integrate two input images. VMDiff ensures structural plausibility and semantic balance through two key components: a *Hybrid Sampling Process (HSP)* and an *Efficient Adaptive Adjustment (EAA)*. HSP integrates the two inputs through noise inversion and feature fusion. The inversion refines an initial noise vector conditioned on a concatenated input object embedding with two parameters and their corresponding text prompt, ensuring deep information mixing to prevent mere juxtaposition. Subsequently, feature fusion employs a curvature-respecting interpolation to blend image embeddings, with a scale factor controlling either object from dominating and thus countering bias generation. EAA automates the search for optimal parameters by proposing a novel similarity-based score that measures alignment with both visual/semantic similarity and balance between the fused object and the input object images/their category labels. By maximizing this score, the EAA dynamically adjusts the influence of each input, ensuring semantically coherent and visually faithful fusions across diverse object pairs.

Our contributions are summarized as follows: **(1)** We introduce a *hybrid sampling process* that constructs optimized semantic noise via guided denoising and inversion, combined with a curvature-aware latent fusion strategy using spherical interpolation for smooth and tunable blending. **(2)** We present an *efficient adaptive adjustment* algorithm that adjusts fusion parameters to achieve semantic and visual balance via a lightweight score-driven search. **(3)** By integrating them, we propose VMDiff, a unified and controllable framework for object-level visual concept fusion. Experiments on a curated benchmark of 780 concept pairs demonstrate that our method achieves superior object synthesis, excelling in semantic consistency, visual harmony, and user-rated creativity.

## 2 RELATED WORK

**Multi-Concept Generation.** Multi-concept generation seeks to synthesize images representing multiple user-defined concepts, typically from a few reference images per concept. Early works such as Custom Diffusion (Kumari et al., 2023) and SVDiff (Han et al., 2023) extend single-concept personalization by fine-tuning on joint data or merging customized models. Later methods (Gu et al., 2023; Liu et al., 2023b) enhance compositionality by merging LoRA modules or token embeddings via gradient fusion (Gu et al., 2023) or spatial inversion (Zhang et al., 2024). More recent approaches further improve efficiency and flexibility: FreeCustom (Ding et al., 2024) employs multi-reference self-attention and weighted masks for training-free composition, while MIP-Adapter (Huang et al., 2025) mitigates object confusion with a weighted-merge strategy. OmniGen (Xiao et al., 2025) and DreamO (Mou et al., 2025) provide unified instruction-based frameworks for diverse generation tasks. Unlike prior methods that explicitly separate input concepts, our approach introduces a unified fusion framework that integrates two concept inputs into a novel object with coherent structure and balanced semantics.

**Semantic Mixing.** Creativity, spanning domains from scientific theories to culinary recipes, has long been a key driver of progress in artificial intelligence (Boden, 2004; Maher, 2010; Wang et al., 2023; Xiong et al., 2025b). In this context, semantic mixing has emerged as a promising approach for generating novel objects by fusing features from multiple concepts into a single coherent representation. Unlike traditional style transfer (Zhang et al., 2023; Tang et al., 2023; Ke et al., 2023) or image editing (Avrahami et al., 2025; Dong & Han, 2023; Brooks et al., 2023; Gal et al., 2023)—which emphasize texture transfer or localized modifications while preserving layout—semantic mixing focuses on concept-level integration within a single entity. Conceptlab (Richardson et al., 2024) interpolates token embeddings to synthesize imaginative entities, while TP2O (Li et al., 2024) enhances controllability by aligning and blending prompt embeddings. However, both operate purely in the textual domain and lack support for real visual content. MagicMix (Liew et al., 2022) fuses image latents with text prompts during denoising, preserving spatial structure, while ATIH (Xiong et al., 2024) improves semantic alignment through more coordinated integration of visual and textual inputs. FreeBlend (Zhou et al., 2025) performs staged interpolation in latent space to produce blended objects. In contrast, our method integrates structural and semantic cues from real image concepts, generating hybrid objects that are both visually coherent and semantically balanced.

## 3 VISUAL MIXING DIFFUSION

In this section, we present a Visual Mixing Diffusion (**VMDiff**) for synthesizing novel objects images in Fig. 3. Our method consists of two key components. We introduce a Hybrid Sampling Process (**HSP, §3.1**) that generates a new object image by blending two distinct inputs using learned scale factors and noise. An Efficient Adaptive Adjustment (**EAA, §3.2**) dynamically adjusts the scale factors and noise based on a Similarity Score (**SS**), ensuring high-quality object synthesis.

### 3.1 HYBRID SAMPLING PROCESS

Given two distinct images $I_1$ and $I_2$, along with their respective category labels $T_1$ and $T_2$ (e.g., *Iron Man* and *Duck*), we first construct a guiding prompt $P_G$: "*A photo of $< T_1 >$ creatively fused with $< T_2 >$.*" and sample an initial Gaussian noise $\epsilon \sim \mathcal{N}(0, I)$. For convenience, we denote an input data $D = \{I_1, I_2, T_1, T_2, P_G\}$. We first employ pretrained image/text encoders $\mathcal{E}_I(\cdot)/\mathcal{E}_T(\cdot)$ of FLUX.1 Krea (Lee et al., 2025) to project both visual and textual modalities into a unified image-language latent space. Specifically, these embeddings are extracted by $z_1 = \mathcal{E}_I(I_1)$, $z_2 = \mathcal{E}_I(I_2)$, $z_p = \mathcal{E}_T(P_G)$. Using these embeddings, HSP includes *blending noise* and *mixing denoise*.

**Blending Noise (BNoise):** Directly sampling standard Gaussian noise to generate a blend of two objects frequently produces incomplete results, with key features such as arms or legs missing (Fig. 4). This occurs because random noise contains no information about the input objects. Our solution is to refine an initial noise vector $\epsilon$, transforming it into a visually and semantically-informed estimate that faithfully represents the source data. Inspired by Rectified Flow (Albergo & Vanden-Eijnden, 2023), this is achieved through a guided denoising and inversion process. Using inputs $\epsilon, z_1, z_2, z_p$,

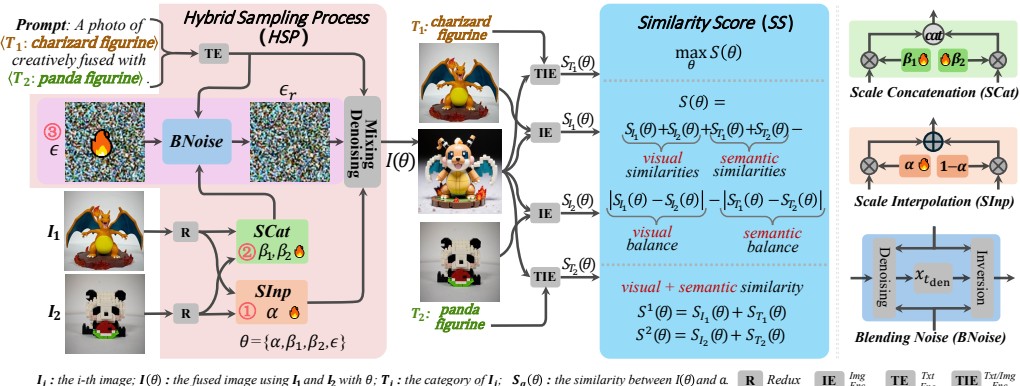

Figure 3: **Overview of our VMDiff framework.** Given two input images and their categories, the Hybrid Sampling Process (HSP) fuses them using noise inversion, scale interpolation (SInp) and scale concatenation (SCat). Efficient adaptive adjustment (EAA) optimizes fusion parameters $\theta = \{\alpha, \beta_1, \beta_2, \epsilon\}$ via a similarity score (SS) that measures visual, semantic, and balance consistency.

we denoise to an intermediate timestep $t_{\text{den}}$, and invert to a refined noise $\epsilon_b$, which is defined as:

$$
\begin{aligned}
\hat{x}_t = x_{t_{\text{den}}} &\Leftarrow \overbrace{x_{t-1} = x_t - (\sigma_t - \sigma_{t-1})v_\phi(x_t, t, z_{\text{SCat}}(z_1, z_2; \beta_1, \beta_2), \gamma_{\text{den}}, z_p)}^{\text{denoise: } t \text{ decreases from } T \text{ to } t_{\text{den}}, \text{ starting } x_T = \epsilon}, \\
\epsilon_b = \underbrace{\hat{x}_T}_{\text{BNoise}} &\Leftarrow \underbrace{\hat{x}_{t+1} = \hat{x}_t + (\sigma_{t+1} - \sigma_t)v_\phi(\hat{x}_t, t, z_{\text{SCat}}(z_1, z_2; \beta_1, \beta_2), \gamma_{\text{inv}}, z_p)}_{\text{inversion: } t \text{ increases from } t_{\text{den}} \text{ to } T, \text{ starting } \hat{x}_t = x_{t_{\text{den}}}},
\end{aligned}
\tag{1}
$$

where $x_t$ and $\hat{x}_t$ are latent variables at timestep $t$, $v_\phi$ denotes the noise prediction network, $\sigma_t$ controls the sampler parameter. For conditioning, we adopt parameters from (Bai et al., 2025): a high denoising strength $\gamma_{\text{den}} = 5$ ensures strong guidance, while an inversion strength of $\gamma_{\text{inv}} = 0$ is used to reduce distortion in the noise space. The total number of timesteps $T$ is 999, with a predefined intermediate denoising timestep at $t_{\text{den}} = 652$. In equation 1, $z_p$ provides the semantic information, while $z_{\text{SCat}}$ provides visual information. Here, we introduce two learnable factors $\beta_1, \beta_2 \in \mathbb{R}_+$ to create a ***scale concatenation (SCat)*** of the input latents: $z_{\text{SCat}}(z_1, z_2; \beta_1, \beta_2) = \text{concat}(\beta_1 z_1, \beta_2 z_2)$.

***Discussion on BNoise: concatenate vs. interpolate.*** We hypothesize that interpolating mismatched embeddings obscures subtle features, while concatenation preserves them, allowing the inversion process to refine noise containing the full concept. To test this, we compare *Interpolate before BNoise:* Blend embeddings first, then refine the noise, and *Interpolate after BNoise:* Refine noise from each embedding first, then blend the results. Fig. 4 shows that both interpolation methods fail to capture intricate details (e.g., legs), whereas our concatenation yields superior visual quality and faithfulness by preserving input details and ensuring a coherent denoising pathway. ***Quantitative results in Appdx. A***.

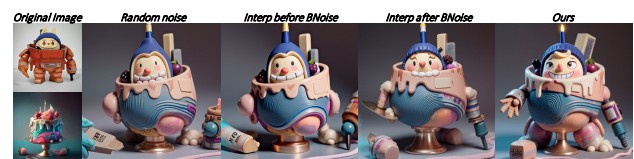

Figure 4: Different BNoise strategies.

**Mixing Denoise (MDeNoise):** Using the blended noise $\epsilon_b$, we denoise it to finally produces a cross-object fusion by mixing the inputs, $z_1, z_2, z_p$. Specifically, we formulate this process as:

$$
I = \mathcal{D}(x_0), \text{ where } x_0 \Leftarrow \overbrace{x_{t-1} = x_t - (\sigma_t - \sigma_{t-1})v_\phi(x_t, t, z_{\text{SInp}}(z_1, z_2; \alpha), \gamma_{\text{gen}}, z_p)}^{\text{MDeNoise: } t \text{ decreases from } T \text{ to } 0, \text{ starting } x_T = \epsilon_b}.
\tag{2}
$$

Here, $\gamma_{\text{gen}} = 4.0$ is a fixed guidance scale, and the decoder $\mathcal{D}(\cdot)$ generate the final fusion image $I$ using the FLUX.1 Krea decoder (Lee et al., 2025). The ***scale interpolation (SInp)***, $z_{\text{SInp}}(z_1, z_2; \alpha)$, mixes the two visual embeddings $z_1$ and $z_2$ into a single coherent representation, which is implemented by a spherical interpolation (Shoemake, 1985): $z_{\text{SInp}}(\alpha) = \frac{\sin(\alpha \cdot \delta)}{\sin(\delta)} z_1 + \frac{\sin((1-\alpha) \cdot \delta)}{\sin(\delta)} z_2$,

where $\delta = \cos^{-1}(z_1 \cdot z_2)$, and $0 \leq \alpha \leq 1$ is a learnable factor to control the mixing ratio. This MDeNoise process in equation 2 outputs the final fusion image $I$.

***Discussion on MDeNoise: interpolate vs. concatenate.*** MDeNoise prioritizes fusing its two inputs, unlike BNoise which preserves them. While concatenation retains more input information, its rigid separation often creates disjointed representations and generations. However, interpolation enables seamless integration. To demonstrate this, we compare with a concatenation-fusion variant: $z_{\text{SInp}}$ is replaced by $z_{\text{SCat}}(\alpha) = \text{concat}(\alpha z_1, (1-\alpha)z_2)$ in equation 2

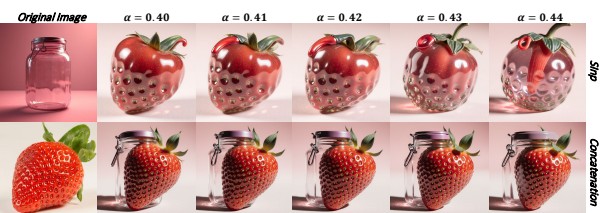

Figure 5: Different MDeNoise generations across $\alpha$.

(Fig. 5), which tends to produce isolated objects rather than a unified hybrid. Our interpolation instead creates a single, coherent entity with harmonious consistency.

**HSP:** Overall, for a given input $D$, the hybrid sampling process combines the BNoise (equation 1) and MDeNoise (equation 2). To simplify the notation, we formalize this process as the function:

$$I(\theta) = \text{HSP}(D; \theta, \hat{\theta}) = \mathcal{D}(x_0), \tag{3}$$

where $\theta = \{\alpha, \beta_1, \beta_2, \epsilon\}$ are learnable parameters, and $\hat{\theta} = \{\gamma_{\text{den}} = 5, \gamma_{\text{inv}} = 0, \gamma_{\text{gen}} = 4, T = 999, t_{\text{den}} = 652\}$ are fixed defaults in this paper.

## 3.2 EFFICIENT ADAPTIVE ADJUSTMENT (EAA)

The HSP process yields distinct fusion results $I(\theta)$ defined in equation 3 with parameters $\theta$, defaults $\hat{\theta}$ and inputs $D$, making parameter selection critical for high-quality synthesis. We propose an adaptive framework to jointly adjust $\theta = \{\alpha, \beta_1, \beta_2, \epsilon\}$, aiming to achieve both semantic coherence and visual fidelity. Inspired by prior work (Li et al., 2024; Xiong et al., 2024), we first introduce a **Similarity Score (SS)** to guide this search: (*For simplicity, input $D$ and defaults $\hat{\theta}$ are not shown.*)

$$S(\theta) = \underbrace{S_{I_1}(\theta) + S_{I_2}(\theta)}_{\text{visual similarity}} + \underbrace{S_{T_1}(\theta) + S_{T_2}(\theta)}_{\text{semantic similarity}} - \underbrace{|S_{I_1}(\theta) - S_{I_2}(\theta)|}_{\text{visual balance}} - \underbrace{|S_{T_1}(\theta) - S_{T_2}(\theta)|}_{\text{semantic balance}}, \tag{4}$$

where $S_{I_i}(\theta)$ $(i = 1, 2)$ is the visual similarity between $I(\theta)$ and the source image $I_i$, computed via a DINO encoder (Oquab et al., 2024), while $S_{T_i}(\theta)$ $(i = 1, 2)$ is the semantic similarity between $I(\theta)$ and the category label $T_i$, measured using CLIP (Radford et al., 2021). This scoring function is designed to optimize two key objectives for successful fusion: (i) *maximizing similarity*, and (ii) *enforcing balance*. The first two terms ensure that the generated image $I(\theta)$ retains high perceptual and semantic fidelity to both input images and their corresponding category labels. By maximizing similarity to both sources, these terms preserve the core features of the original concepts. The final two terms—penalizing the absolute differences—explicitly enforce *balance*, preventing the model from overfitting to one input and encouraging a fair integration of both objects' features. Together, these components create a unified SS objective that balances fidelity and symmetry, offering a principled framework for optimizing feature fusion parameters.

**Our EAA Algorithm.** To maximize this objective $S(\theta)$ in equation 4, we present a hierarchical adjustment strategy that learns the parameters $\theta = \{\alpha, \beta_1, \beta_2, \epsilon\}$ using the acceptance threshold $Th = 2.4$. The key loop iterates from $k = 1$ to $K = 3$, performing these steps:

    ① **Sample (initial) Gaussian noise:** $\epsilon \sim \mathcal{N}(0, I)$, **initialize the parameters:** $\alpha = 0.5, \beta_1 = \beta_2 = 1.0$.

    ② **Searching $\alpha$:** Fixed $\beta_1 = \beta_2 = 1.0$ and $\epsilon$, perform a golden section search (Teukolsky et al., 1992) to find the optimal mixing factor $\alpha^*$:

$$\alpha^* = \arg\max_{\alpha \in [0,1]} S(\alpha, \beta_1, \beta_2, \epsilon). \tag{5}$$

③ **Adjusting** $\beta_1, \beta_2$**:** Fixed $\alpha^*, \epsilon$, if $S(\alpha^*, \beta_1, \beta_2, \epsilon) \leq Th$, then update the noise factors:

$$\begin{cases} \beta_1^* = \beta_1 \ \& \ \beta_2^* = \arg\max_{\beta_2 \in \mathbb{R}_+} S(\alpha^*, \beta_1, \beta_2, \epsilon), & \text{if } S_1 > S_2, \\ \beta_2^* = \beta_2 \ \& \ \beta_1^* = \arg\max_{\beta_1 \in \mathbb{R}_+} S(\alpha^*, \beta_1, \beta_2, \epsilon), & \text{otherwise.} \end{cases} \tag{6}$$

where $S_1 = S_{I_1} + S_{T_1}$, $S_2 = S_{I_2} + S_{T_2}$, and $S_1 > S_2$ indicates that the mixing noise favors the object $I_1$, and vice versa.

④ **Acceptance criterion:**

$$\begin{cases} \epsilon^* = \epsilon \ \& \ \textbf{return } \theta^* = \{\alpha^*, \beta_1^*, \beta_2^*, \epsilon^*\}, & \text{if } S(\alpha^*, \beta_1^*, \beta_2^*, \epsilon) > Th, \\ \textbf{return } \theta^* = \{\alpha^*, \beta_1^*, \beta_2^*, \epsilon^*\} \ \& \ \textbf{break}, & \text{if } k > K, \\ \textbf{turn to the step } ① \ \textbf{to resample } \epsilon \ \& \ k{+}{+}, & \text{otherwise.} \end{cases} \tag{7}$$

where the fused object image $I(\theta)$ is defined in equation 3. Our adaptive loop efficiently explores a low-dimensional yet expressive parameter space $\theta = \{\alpha, \beta_1, \beta_2, \epsilon\}$, yielding conceptually balanced and perceptually smooth fusion results (Fig. 9). By reusing intermediate predictions and limiting optimization to scalar-level searches (via golden section search), the method enhances sample efficiency—avoiding the computational overhead of gradient-based latent-space backpropagation.

***Discussion on resampling*** $\epsilon$***.*** During our blending process, sampling random Gaussian noise can occasionally yield low-quality or failed fusions. While first-order optimization is an intuitive solution, it offers no significant advantage over simple zero-order resampling for diffusion generation, despite its higher cost (Ma et al., 2025). Consequently, we adopt a zero-order resampling strategy to search for $\epsilon$, and a small number of resamples $K = 3$ proves sufficient for high-quality fusion. ***For fair comparison, this resampling is disabled,*** $K = 1$***, and the random seed is fixed at 42.***

## 4 EXPERIMENTS

### 4.1 EXPERIMENTAL SETTINGS

**Datasets.** We introduce IIOF (Image-Image Object Fusion), a new benchmark of 780 image pairs derived from 40 objects across four classes (i.e., animals, fruits, artificial objects, and character figurines). Most images are from PIE-Bench (Ju et al., 2024) and Pexels[3]; figurines were self-captured for quality. To evaluate order-sensitive methods, we also generate all ordered pairs (1,560 total), ensuring a comprehensive and fair benchmark. ***More details in Appdx. B.***

**Implementation Details.** Our method builds upon FLUX.1 Krea (Lee et al., 2025), implementing $\mathcal{E}_I$ with Redux (Black Forest Labs, 2024) for latent-space alignment. We generate all images at $512\times 512$ resolution using the FlowMatchEulerDiscreteScheduler (Lipman et al., 2022) with 20 denoising steps. For the Efficient Adaptive Adjustment (EAA) module, we use Grounded-SAM (Ren et al., 2024) and the query *"most prominent object"* to localize main regions for visual and semantic similarity computation. Each parameter search for $\alpha$ and $\beta$ involves at most 10 image generations. All experiments are conducted on two NVIDIA RTX 4090 GPUs.

**Evaluation Metrics.** To evaluate our method, we use two metric families: Semantic Alignment (SA) and Single-entity Coherence (SCE). SA is computed on the generated prompt $P_G$ using VQAScore (Lin et al., 2024b) and LLaVA-Critic (Xiong et al., 2025a). VQAScore employs CLIP-FlanT5 (Roberts et al., 2022) and LLaVA (Liu et al., 2023a), denoted as $\text{VQA}_{\text{T5}}^{\text{SA}}$ and $\text{VQA}_{\text{LLaVA}}^{\text{SA}}$, respectively; the LLaVA-Critic score is $\text{LC}^{\text{SA}}$. SCE assesses if the image forms a unified concept by asking: *"A photo of a seamless fusion of $<T_1>$ and $<T_2>$ into a single coherent entity."* Its scores are $\text{VQA}_{\text{T5}}^{\text{SCE}}$, $\text{VQA}_{\text{LLaVA}}^{\text{SCE}}$, and $\text{LC}^{\text{SCE}}$. We also compute the SS score and the balance metric $B_{\text{sim}} = |S_{I_1}(\theta) - S_{I_2}(\theta)| + |S_{T_1}(\theta) - S_{T_2}(\theta)|$, where $S_{T_i}(\theta)$ are normalized to $[0, 1]$ using empirical bounds 0.15 and 0.45 to align the scales of visual and textual modalities.

### 4.2 MAIN RESULTS

We compare with leading methods across three categories: (i) multi-concept generation (e.g., OmniGen (Xiao et al., 2025), FreeCustom (Ding et al., 2024), MIP-Adapter (Huang et al.,

---

[3]https://www.pexels.com/

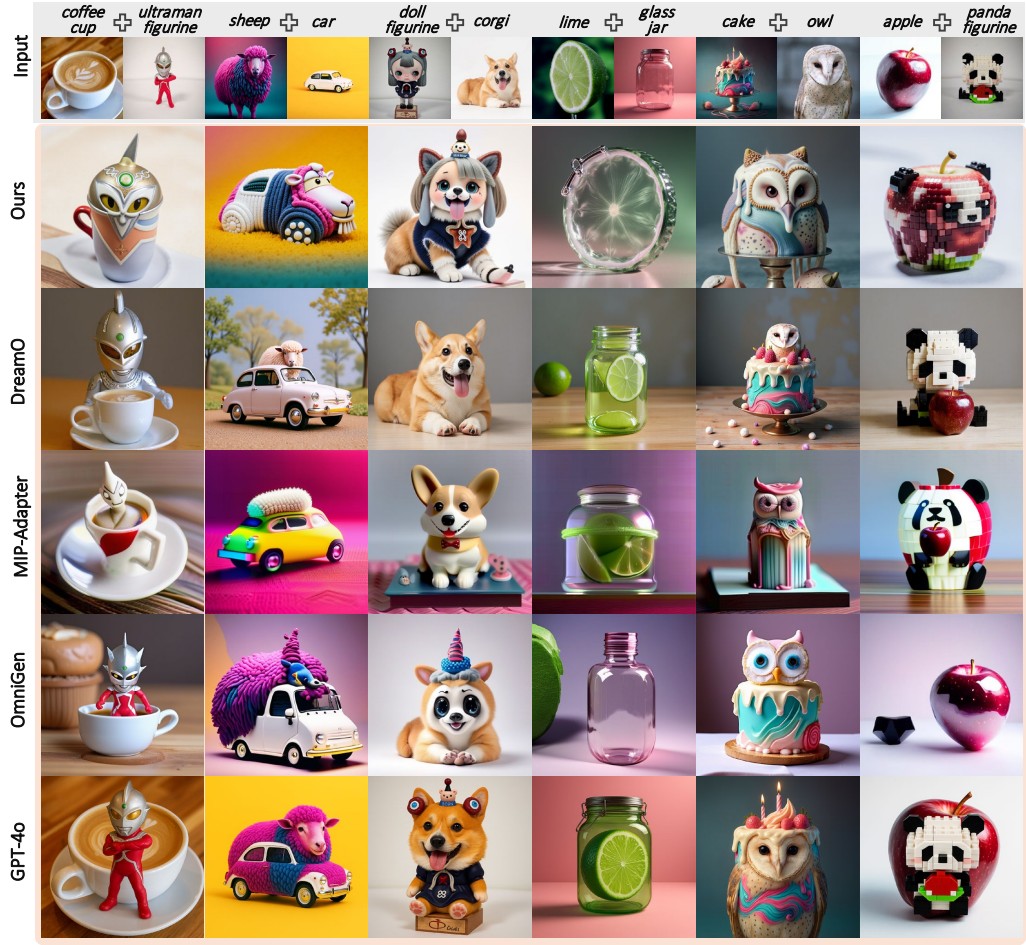

Figure 6: **Comparisons with Multi-Concept Generation Methods.** Our approach yields hybrid objects with improved structural coherence and visual balance over existing methods.

2025), DreamO (Mou et al., 2025)), (ii) mixing-based (e.g., ATIH (Xiong et al., 2024), Conceptlab (Richardson et al., 2024), FreeBlend (Zhou et al., 2025)), and (iii) image editing (e.g., Stable Flow (Avrahami et al., 2025)). We also include qualitative results from GPT-4o (OpenAI, 2025). Inputs vary: multi-concept methods use two images and a text prompt; ATIH and Stable Flow use one image and text; Conceptlab uses text only. ***More examples in Appdx. G.***

**Qualitative Comparison.** Fig. 6 compares our method with multi-concept generation baselines (e.g., MIP-Adapter, OmniGen, DreamO, GPT-4o), highlighting two observations. First, baselines output often merely overlay features rather than fusing them—for example, *a lime enclosed in a glass jar without integration*—while our method creates a coherent hybrid. Second, baselines frequently favor one concept, such as generating either a doll or a corgi but not a unified blend. In contrast, our approach balances both concepts, producing structurally unified and semantically consistent results. This demonstrates our method's superior ability to achieve fine-grained visual fusion.

Fig. 7 qualitatively compares our method with mixing/editing baselines (e.g., Conceptlab, ATIH, FreeBlend, Stable Flow). Conceptlab often biases toward one concept, while Stable Flow and ATIH make only subtle edits, such as color or texture transfer. FreeBlend frequently loses original information and yields fragmented outputs. In contrast, our approach synthesizes novel objects that structurally and visually integrate both concepts, achieving a deeper, more harmonious fusion and demonstrating superior blending capability.

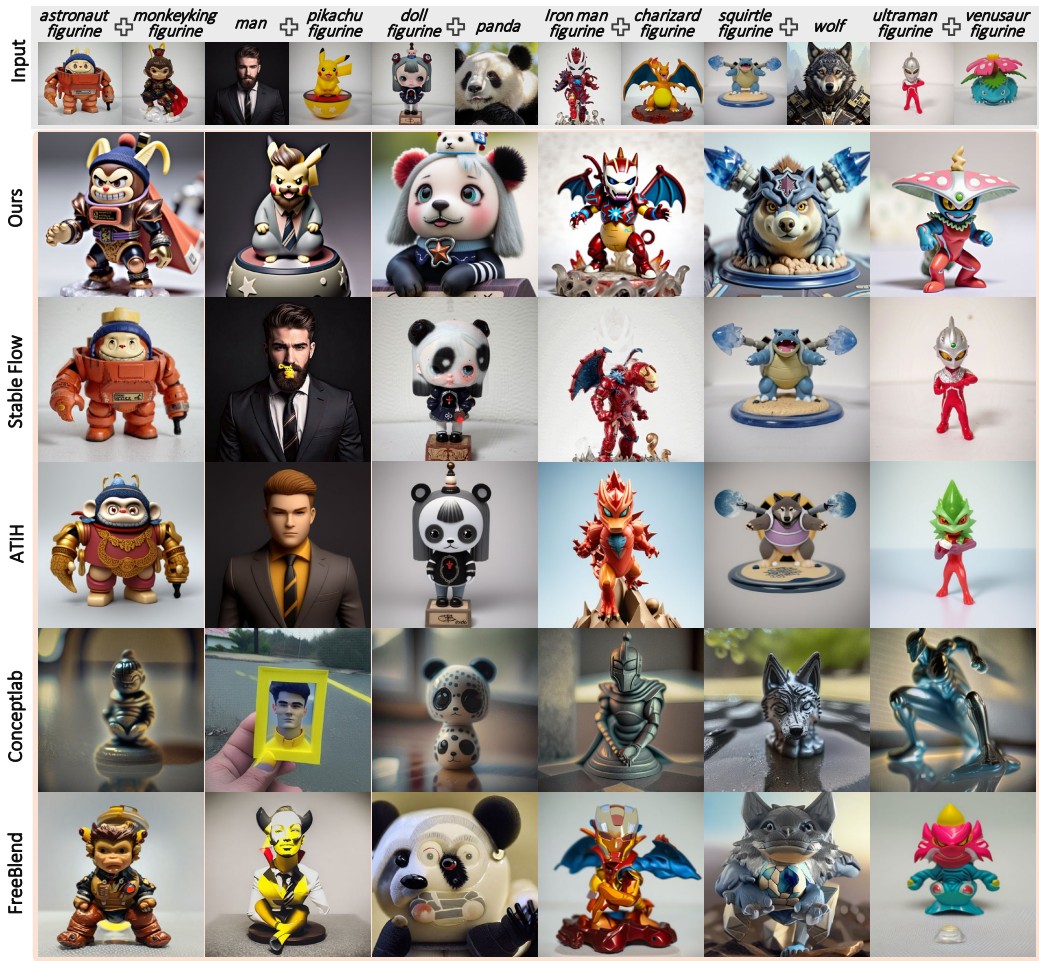

Figure 7: **Comparisons with Mixing and Image Editing Methods.** Our method produces more coherent and balanced hybrids, while baselines often favor one concept or apply minimal edits.

Table 1: Quantitative comparisons on our IIOF dataset.

| Models | $\text{VQA}^{SA}_{T5}\uparrow$ | $\text{VQA}^{SCE}_{T5}\uparrow$ | $\text{LC}^{SA}\uparrow$ | $\text{LC}^{SCE}\uparrow$ | $\text{VQA}^{SA}_{LLaVA}\uparrow$ | $\text{VQA}^{SCE}_{LLaVA}\uparrow$ | $SS\uparrow$ | $B\text{sim}\downarrow$ |
|---|---|---|---|---|---|---|---|---|
| **Our VMDiff** | 0.639 | 0.540 | 8.372 | 8.392 | 0.390 | 0.413 | 2.068 | 0.324 |
| FreeCustom (CVPR (Ding et al., 2024)) | 0.579 | 0.452 | 6.958 | 6.946 | 0.360 | 0.388 | 1.580 | 0.776 |
| MIP-Adapter (AAAI (Huang et al., 2025)) | 0.621 | 0.512 | 8.301 | 8.076 | 0.389 | 0.417 | 1.866 | 0.483 |
| OmniGen (CVPR (Xiao et al., 2025)) | 0.570 | 0.469 | 7.550 | 7.233 | 0.352 | 0.348 | 1.705 | 0.617 |
| Conceptlab (TOG (Richardson et al., 2024)) | 0.573 | 0.483 | 7.589 | 7.728 | 0.362 | 0.395 | – | – |
| ATIH (NeurIPS (Xiong et al., 2024) ) | 0.523 | 0.465 | 7.275 | 6.816 | 0.317 | 0.367 | – | – |
| Stable Flow (CVPR (Avrahami et al., 2025)) | 0.460 | 0.372 | 6.020 | 5.024 | 0.266 | 0.294 | – | – |
| DreamO (SIGGRAPH Asia (Mou et al., 2025) ) | 0.591 | 0.467 | 7.592 | 7.013 | 0.370 | 0.346 | 1.793 | 0.644 |
| FreeBlend (arXiv (Zhou et al., 2025)) | 0.588 | 0.507 | 7.836 | 7.788 | 0.341 | 0.383 | 1.870 | 0.479 |

**Quantitative Comparison.** Table 1 presents quantitative comparisons on key metrics, including $\text{VQA}^{SA}_{T5}$, $\text{VQA}^{SA}_{LLaVA}$, $\text{VQA}^{SCE}_{T5}$, $\text{VQA}^{SCE}_{LLaVA}$, $\text{LC}^{SA}$, $\text{LC}^{SCE}$, similarity score (SS), and fusion balance $B_{sim}$. Although MIP attains the highest $\text{VQA}^{SCE}_{LLaVA}$, it ranks only second or below on the other VQA, LC, SS, and $B_{sim}$ metrics, indicating that its improvements are not holistic. In contrast, our method consistently outperforms all baselines on most met-

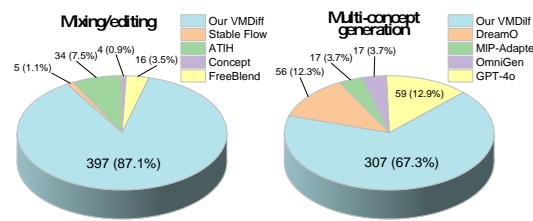

Figure 8: User studies.

Table 2: Quantitative ablation study on our IIOF dataset.

| Models | VQA$_{T5}^{SA}\uparrow$ | VQA$_{T5}^{SCE}\uparrow$ | LC$^{SA}\uparrow$ | LC$^{SCE}\uparrow$ | VQA$_{LLaVA}^{SA}\uparrow$ | VQA$_{LLaVA}^{SCE}\uparrow$ | $SS\uparrow$ | $B$sim$\downarrow$ |
|---|---|---|---|---|---|---|---|---|
| Baseline 1 | 0.497 | 0.438 | 7.261 | 7.077 | 0.287 | 0.314 | 1.570 | 0.682 |
| Baseline 2 | 0.508 | 0.441 | 7.426 | 7.291 | 0.298 | 0.325 | 1.586 | 0.693 |
| Baseline 2+$\alpha$-search | 0.625 | 0.532 | 8.278 | 8.276 | 0.382 | 0.405 | 2.025 | 0.358 |
| Baseline 2+$\alpha$-search+$\beta_1,\beta_2$-search | **0.639** | **0.540** | **8.372** | **8.392** | **0.390** | **0.413** | **2.068** | **0.324** |

rics, demonstrating strong capability in generating coherent and natural blended objects.

These results reinforce our qualitative findings and confirm the effectiveness of our approach in achieving high-quality visual fusion.

**User Study.** To evaluate the perceptual quality of our fusions, we conducted two user studies (Fig. 8). 76 participants each rated 12 results—6 from *Multi-Concept Generation* and 6 from *Mixing/Editing*—yielding 912 total votes. Our VMDiff received the highest preference in both groups: **67.3%** and **87.1%**, respectively. GPT-4o and ATIH ranked second, but with significantly lower votes (12.9% and 7.5%). These results indicate that our VMDiff aligns better with human preferences in visual coherence and creativity. ***More details in Appdx. C.***

## 4.3 ABLATION STUDY

We conducted an ablation study to evaluate the contributions of our VMDiff's key components, as shown in Fig. 9 and Table 2. Progressively adding each element—**(i)** *baseline 1:* random noise+MDeNoise ($\alpha = 0.5$), **(ii)** *baseline 2:* baseline 1+BNoise ($\beta_1 = \beta_2 = 1$), **(iii)** baseline 2 + MDeNoise ($\alpha$ search), and **(iv)** baseline 2 + BNoise ($\beta_1, \beta_2$ search) + MDeNoise ($\alpha$ search)—yielded consistent improvements. Without noise refinement, outputs lacked detail. Its

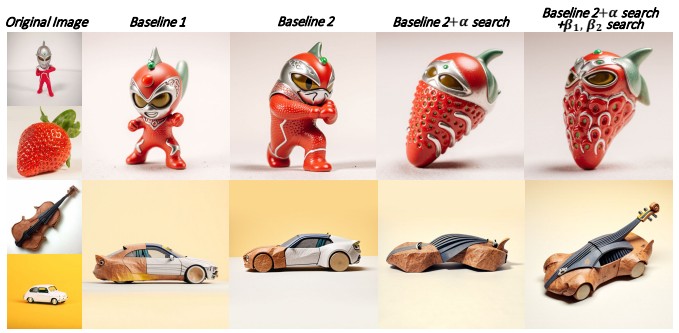

Figure 9: **Ablation study in VMDiff.** *Noise refinement* improves detail and structure, while *adaptive $\alpha$ and $\beta$ search* progressively enhance semantic balance and visual coherence.

inclusion enhanced structural fidelity and preserved input features. Adaptive $\alpha$ improved fusion balance, while adaptive $\beta$ refined noise influence for greater visual harmony. Fig. 10 illustrates the optimization process for a representative case (*doll figurine + rabbit*). Throughout iterations, similarity $S(\theta)$ (green) increased steadily, while the blending balance metric (dark blue) decreased. The $\alpha$ search (light blue) rapidly boosted similarity, and $\beta$ search (orange) smoothed visual-textual alignment. These results confirm that our EAA design effectively optimizes both similarity and symmetry for high-quality blending. ***Limitations are discussed in Appdx. D.***

## 4.4 MULTI-IMAGE FUSION.

**Multi-image fusion.** We also explore extending VMDiff beyond pairwise fusion. Figure 11 shows preliminary three-image results obtained by sequentially applying our pipeline (e.g., first fusing $(I_1, I_2)$ and then fusing the hybrid with $I_3$). The method can still produce single coherent entities that blend attributes from all three categories, indicating that our formulation can, in principle, scale to more inputs. However, compared with the pairwise case, these hybrids exhibit stronger information loss and imbalance across sources, so in this work we focus on image pairs and leave permutation-invariant, learned aggregation of multiple image embeddings to future work.

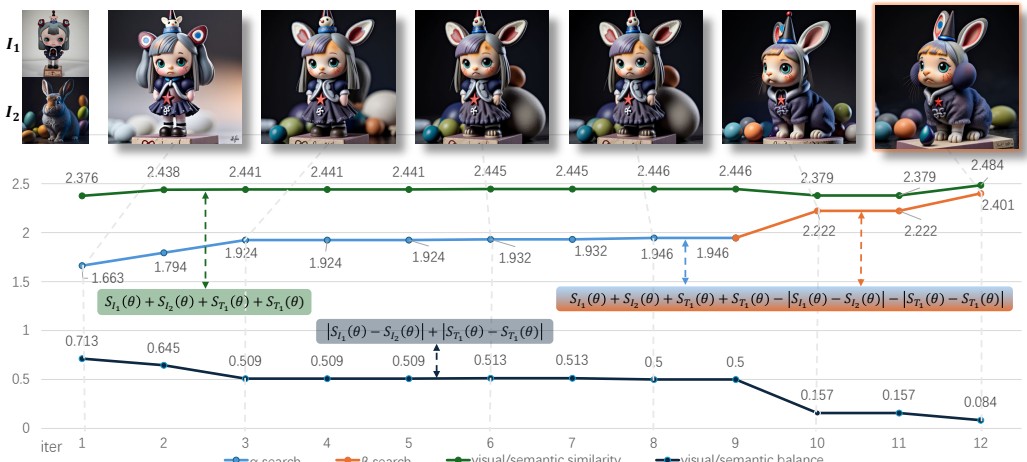

Figure 10: **Visualizing the updated process of our EAA** based on two input images $I_1$ (*doll figurine*) and $I_2$ (*rabbit*). The $\alpha$ parameter (blue) improves fusion quality, while $\beta$ (orange) enhances semantic balance. The green curve (similarity) rises and the dark blue curve (imbalance) falls over iterations. The final output is a coherent hybrid with high similarity and minimal imbalance.

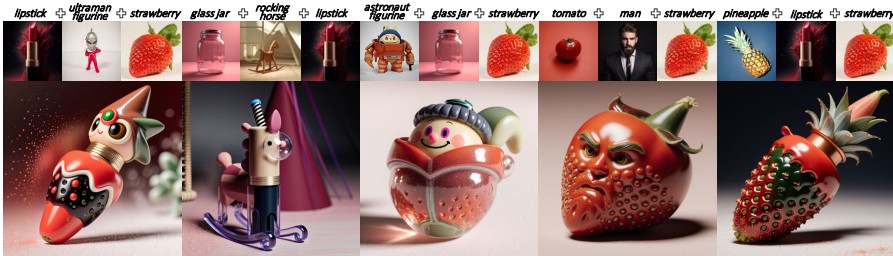

Figure 11: **Multi-image fusion.** Our method can sequentially fuse three images into a single coherent entity, though this results in greater information loss compared to pairwise fusion.

## 5 CONCLUSION

In this paper, we presented VMDiff, a novel unified and controllable framework for visual concept fusion that synthesizes coherent new objects directly from two input images. Our approach enables fine-grained control by semantically integrating concepts at both the noise and latent levels. VMDiff consists of two core components: (1) a hybrid sampling process that constructs optimized semantic noise through guided denoising and inversion, followed by a curvature-aware latent fusion using spherical interpolation, and (2) an efficient adaptive adjustment algorithm that refines fusion parameters via a lightweight, score-driven search. Experimental results on a curated benchmark demonstrate VMDiff's superior performance, excelling in semantic consistency, visual harmony, and user-rated creativity, thereby establishing a new paradigm for hybrid object synthesis. This work offers practical and valuable insights for professionals developing combinational characters, directly applicable to diverse fields from film and animation to figures and industrial design.

## 6 ACKNOWLEDGMENTS

This work was supported by National Natural Science Fund of China (Nos. U24A20330, 62361166670, 62506155), Provincial Natural Science Fund of Jiangsu (Nos. BK20251985), and Suzhou Municipal Leading Talents Fund (Nos. ZXL2025320).

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
