## SUPPLEMENTARY MATERIALS

This supplementary material provides additional technical details and extended results to support the main paper. We begin in **Section A** with two key discussions: the necessity of adjusting $\beta_1$ and $\beta_2$ in our hierarchical parameter search, and a quantitative comparison of BNoise fusion strategies—concatenation versus interpolation. **Section B** describes the construction of our proposed IIOF benchmark dataset, including the criteria for category selection and object pairing strategies. **Section C** presents a comprehensive user study, providing human preference validation of our fusion results. In **Section D**, we outline the current limitations of our method, discuss remaining challenges, and suggest possible directions for future improvement. **Section E** contains our formal statement on the use of LLMs in this work, in accordance with ICLR policy. **Section F** details the full inference pipeline of our VMDiff framework. Finally, **Section G** showcases extensive qualitative results, further demonstrating the effectiveness and generalization ability of our method across diverse fusion scenarios.

## A  ADDITIONAL DISCUSSIONS

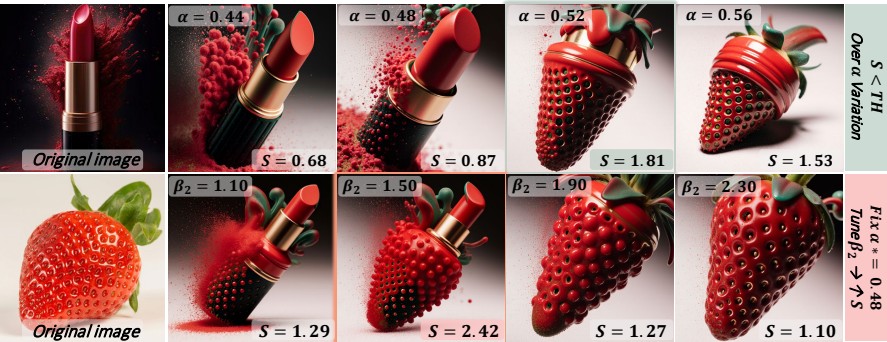

Figure 12: Illustration of our hierarchical parameter adjustment. The top row shows results from searching $\alpha$; the bottom row refines the fusion by fixing $\alpha$ and adjusting $\beta_2$. **Consistent with Sec. 3.2, once the overall score $S$ exceeds the acceptance threshold $T_h = 2.4$, the fusion becomes visually coherent and balanced**; when $\alpha$-only optimization underperforms, the second-stage $\beta_2$ refinement raises $S$ above the threshold.

***Discussion on the necessity of adjusting $\beta_1, \beta_2$.*** As shown in Fig. 12, global optimization over $\alpha$ alone occasionally fails to yield well-fused results. To mitigate this, we first fix $\alpha^*$ (corresponding to the best similarity score in Eq. 4) and then perform a local refinement by optimizing $\beta_1, \beta_2$. This adjustment allows the model to precisely calibrate the noise contribution of each object, enhancing both visual coherence and semantic balance in the final output.

***Discussion on BNoise.*** As shown in Table 3 on the IIOF dataset, Ours (Concat before inversion) achieves state-of-the-art performance on most metrics. Although it ranks second on the LC metric, its substantial advantage on SS demonstrates that concatenation more effectively preserves and integrates complementary information from both inputs. In summary, concatenation before inversion yields superior visual quality and semantic faithfulness by retaining fine-grained details and guiding a more coherent denoising pathway, compared with either form of interpolation.

Table 3: Quantitative Evaluation of BNoise Fusion: Concatenation vs. Interpolation.

| Models | $VQA_{T5}^{SA}\uparrow$ | $VQA_{T5}^{SCE}\uparrow$ | $LC^{SA}\uparrow$ | $LC^{SCE}\uparrow$ | $VQA_{LLaVA}^{SA}\uparrow$ | $VQA_{LLaVA}^{SCE}\uparrow$ | $SS\uparrow$ | $Bsim\downarrow$ |
|---|---|---|---|---|---|---|---|---|
| Random noise | 0.497 | 0.438 | 7.261 | 7.077 | 0.287 | 0.314 | 1.570 | 0.682 |
| Interp Before Inversion | 0.504 | 0.441 | **7.439** | **7.390** | 0.293 | 0.321 | 1.551 | **0.678** |
| Interp After Inversion | 0.486 | 0.430 | 7.278 | 7.112 | 0.283 | 0.311 | 1.532 | 0.712 |
| **Ours(Concat Before Inversion)** | **0.508** | **0.442** | 7.426 | 7.291 | **0.298** | **0.325** | **1.586** | 0.693 |

***Discussion on additional ablation of BNoise and the $\alpha/\beta$ search.*** To better understand the contributions of BNoise and the EAA search, we conduct an additional ablation on 1,184 pairs from IIOF, summarized in Fig. 13 and Table 4. We compare four variants: (i) **Baseline 1**, which uses random noise plus MDeNoise with a fixed $\alpha = 0.5$ (no BNoise, no search); (ii) **Baseline 2**, which augments Baseline 1 with BNoise by setting $\beta_1 = \beta_2 = 1$ (semantic noise injected, no search); (iii) **Baseline 1**

+ $\beta_1, \beta_2$-**search**, which augments Baseline 1 with BNoise and an EAA search over $(\beta_1, \beta_2)$; and (iv) **Random noise** + $\alpha$-**search**, which applies EAA only to $\alpha$ without BNoise.

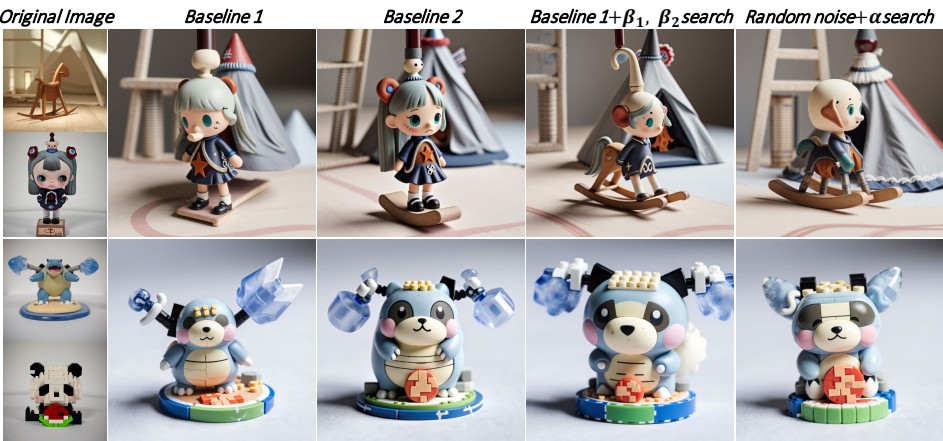

Figure 13: **Ablation of BNoise and the $\alpha/\beta$ search. Each column shows the original image pair (left) and fused results from different variants:** *Baseline 1* (random noise + MDeNoise with fixed $\alpha=0.5$), *Baseline 2* (Baseline 1 + BNoise with $\beta_1=\beta_2=1$), *Baseline 1 + $\beta$-search*, and *Random noise + $\alpha$-search*. **BNoise (columns 2–3) provides a more informative initialization that preserves structures from both sources, while $\beta$-search further balances semantic content in the noise. In contrast, random-noise + $\alpha$-search alone often loses details, confirming the complementary roles of BNoise and the $\alpha/\beta$ search.**

Table 4: Quantitative ablation of BNoise and the $\alpha/\beta$ search on the IIOF dataset.

| Models | $VQA_{T5}^{SA}\uparrow$ | $VQA_{T5}^{SCE}\uparrow$ | $LC^{SA}\uparrow$ | $LC^{SCE}\uparrow$ | $VQA_{LLaVA}^{SA}\uparrow$ | $VQA_{LLaVA}^{SCE}\uparrow$ | $SS\uparrow$ | $Bsim\downarrow$ |
|---|---|---|---|---|---|---|---|---|
| Baseline 1 | 0.496 | 0.419 | 7.186 | 7.065 | 0.283 | 0.320 | 1.563 | 0.691 |
| Baseline 2 | 0.503 | 0.420 | 7.326 | 7.191 | 0.290 | 0.326 | 1.580 | 0.705 |
| Baseline 1+$\beta_1, \beta_2$-search | 0.553 | 0.461 | 7.723 | 7.679 | 0.320 | 0.359 | 1.760 | 0.553 |
| Random noise+$\alpha$-search | 0.603 | 0.508 | 8.009 | 8.017 | 0.357 | 0.394 | 1.972 | 0.354 |

Qualitatively, Baseline 1 often loses information from the sources, whereas Baseline 2 preserves more structures from both inputs, confirming that the semantic noise $\epsilon_b$ obtained via the denoise–invert cycle provides a more informative initialization than pure Gaussian noise. Adding $\beta$-search on top of BNoise further improves all SA/SCE and SS scores and reduces the imbalance metric $B_{\text{sim}}$ from 0.691 to 0.553, indicating that $(\beta_1, \beta_2)$ effectively rebalance how each source contributes to the noise. The random-noise+$\alpha$-search variant achieves higher SA/SCE and lower $B_{\text{sim}}$ than Baseline 1, but still misses fine details and parts from the inputs, as seen in Fig. 13, due to the lack of a semantically informed noise initialization. Taken together, these results highlight complementary roles: BNoise produces a conditional, information-carrying noise $\epsilon_b$, while the EAA search over $\alpha$ and $(\beta_1, \beta_2)$ adjusts the contributions of the two sources in the mixed embeddings and in the noise, respectively. This motivates our full HSP+EAA design, which combines both components to obtain the most faithful and balanced hybrids.

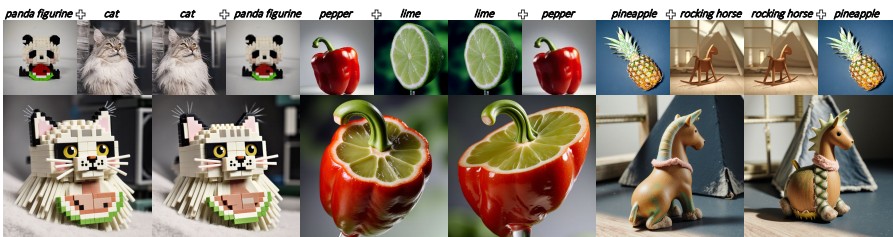

Figure 14: **Effect of swapping the order of $T_1$ and $T_2$ in the prompt.**

***Discussion on name order.*** Fusion is, in principle, sensitive to the order of the category names in the guiding prompt, since the text encoder need not be strictly commutative. To probe this effect, we fix the image pair $(I_1, I_2)$ and all hyperparameters, and only swap the order of the category tokens $T_1$

and $T_2$ in the prompt, using the "Random noise + MDeNoise ($\alpha = 0.5$)" baseline (Fig. 14). For most pairs (left and middle examples), the two orderings produce almost identical hybrids, indicating that our fusion behaves approximately symmetric with respect to name order. In a few harder cases (right), the leading token receives slightly more emphasis and extra attributes may appear, but both generations remain single, coherent hybrids rather than collapsing to one source. In the full VMDiff pipeline, this mild asymmetry is further reduced by the symmetric fusion score $S(\theta)$ and EAA search, which explicitly discourage strong bias toward a single category.

***Discussion on fusion strategy.*** To better understand why we favor interpolation over concatenation, we also test a weighted concatenation variant $z_{\text{cat}}(\alpha) = \text{concat}(\alpha z_1, (1-\alpha)z_2)$, and fix the source images while varying $\alpha$ from 0.1 to 0.8 (Fig. 15). As the figure shows, relatively large changes in $\alpha$ are required to noticeably alter the result, confirming that $\alpha$ exerts much weaker control in the concatenation space than in the interpolated space. More importantly, across all settings the fusion remains *stitching-like*: one region of the image is dominated by the strawberry and the other by the jar, with a clear boundary between them. This suggests that separating $z_1$ and $z_2$ into distinct blocks encourages the network to treat them as two pieces to be glued together, rather than a single coherent object. In contrast, our MDeNoise stage mixes $z_1$ and $z_2$ via spherical interpolation within the *same* latent subspace, leading to much more integrated hybrids with smoothly shared geometry and appearance (see Fig. 21). These observations support our choice of slerp-based mixing in MDeNoise rather than concatenation-based fusion.

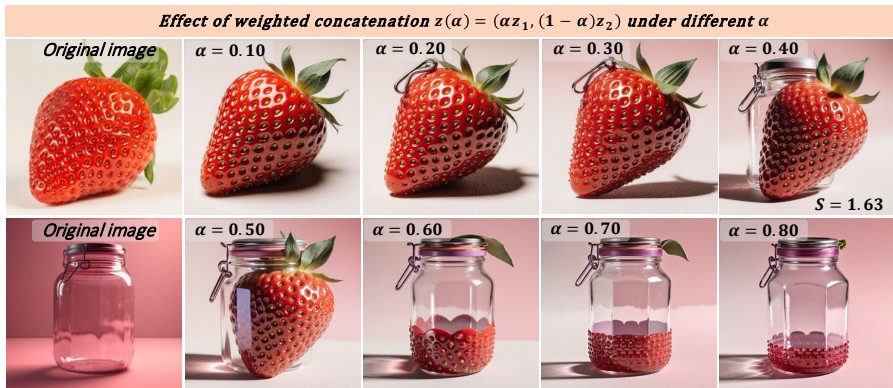

Figure 15: **Behaviour of weighted concatenation** $z_{\text{cat}}(\alpha) = \text{concat}(\alpha z_1, (1-\alpha)z_2)$ **under different** $\alpha$**.** We fix the source images and vary $\alpha$ from 0.1 to 0.8. Large changes in $\alpha$ are required to noticeably alter the result, and across all settings the fusion remains stitching-like: one region is dominated by the strawberry and the other by the jar, with a clear boundary between them.

***Why MIP-Adapter scores higher on SA/SCE.*** At first glance, MIP-Adapter appears visually weaker than DreamO and OmniGen, yet it achieves higher SA/SCE scores in Table 1. This is because our metrics are explicitly designed to measure *semantic fusion quality* rather than photo-realism. SA and SCE are LMM-based scores that reward (i) strong alignment with the fusion prompt and (ii) the presence of a *single* fused entity that simultaneously reflects both source categories. As illustrated in Fig. 16, MIP-Adapter typically produces one coherent object that clearly contains cues from both inputs, even though many fine-grained instance details are washed out. DreamO and OmniGen, on the other hand, often generate highly realistic and aesthetically pleasing images, but they frequently either omit one concept or render two separate objects instead of a single hybrid. Such behaviours are explicitly penalized by SA/SCE (and SS), which explains why MIP-Adapter scores higher in Table 1 despite being less visually appealing than DreamO and OmniGen in Fig. 16 and receiving lower user preference in Table 6.

***Backbone generalization.*** We evaluate VMDiff on three backbones with identical settings: Flux-dev+Redux, SDXL (Lin et al., 2024a)+IP-Adapter (Ye et al., 2023), and SD-3.5 (AI, 2024)+SD-3.5-IP-Adapter (Team, 2024) (Fig. 17). All three run under our HSP+EAA framework, so VMDiff is not tied to FLUX.1 Krea, but the *quality and tendency toward a single hybrid object* strongly depend on how image information is encoded. Flux+Redux maps images into a semantic latent space shared with text, allowing BNoise+SInp to operate directly on rich, text-like image embeddings and thus best preserve instance-level geometry and appearance from both sources. For SDXL and

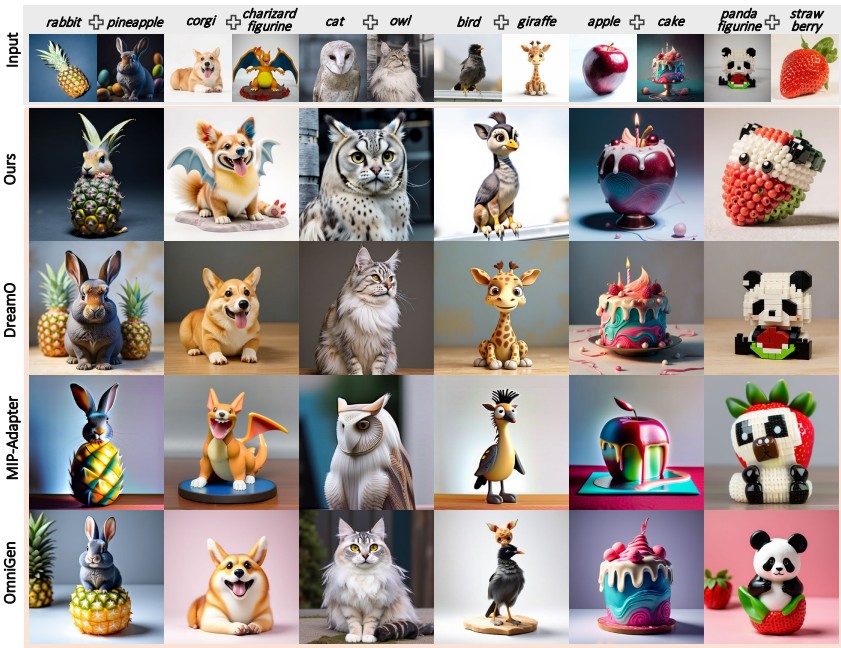

Figure 16: **Additional qualitative comparisons on IIOF.**

especially SD-3.5, IP-Adapter injects image features as extra attention tokens; interpolating these tokens mainly modulates high-level semantics and, in our results, often weakens retention of input-specific structure. VMDiff therefore benefits most from backbones that preserve detailed instance information in a text-compatible embedding space. Our SDXL and SD-3.5 experiments should be viewed as feasibility checks under this weaker image interface, and we expect that adding Redux-style semantic image encoders to such models would narrow the quality gap to FLUX.1 Krea.

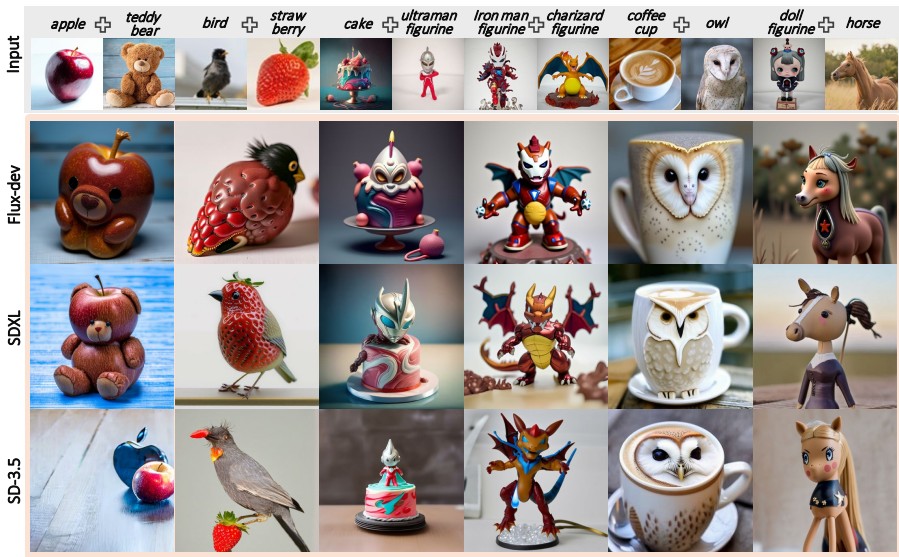

Figure 17: **Each column shows an input pair (top) and the fused outputs when plugging our framework into Flux-1.0-dev+Redux, SDXL+IP-adapter(Ye et al., 2023), and SD-3.5+SD-3.5-IP-adapter(Team, 2024). All backbones use the same images and fusion prompt.**

## B  DATASETS

To systematically evaluate our fusion framework, we construct a comprehensive benchmark dataset named **IIOF** (Image-Image Object Fusion), specifically tailored for assessing diverse and semantically rich visual concept mixing.

We meticulously selected **40 distinct object categories**, strategically organized into four semantic groups: *Animals*, *Fruits*, *Artificial Objects*, and *Character Figurines*. Each group comprises 10 unique classes, a design choice that ensures both intra-group consistency and ample inter-group diversity. A complete list of all selected categories is provided in Table 5.

For each chosen class, we sourced one high-quality, representative image. The majority of these images were obtained from established public benchmarks such as PIE-Bench (Ju et al., 2024) and popular stock image platforms like Pexels[4]. Recognizing the scarcity of high-quality, publicly available data for character figurines, we self-captured these images under controlled conditions, ensuring consistent lighting and resolution to maintain visual quality and diversity across the dataset. Figure 18 showcases all the selected images, providing a visual overview of the dataset's content. Additionally, each selected image is paired with its corresponding **textual category name**, as detailed in Table 5, to facilitate evaluations for prompt-based fusion methods.

Initially, we derived **780 unique image pairs** by combining each of the 40 objects with every other object once, without considering input order. However, to ensure a comprehensive evaluation and enable fair comparison across all methods, particularly those sensitive to input order (e.g., ATIH (Xiong et al., 2024)), we further expanded IIOF to include **all possible ordered pairs** among the 40 categories. This expansion yielded a total of **1,560 image pairs**, where each combination $(A, B)$ is present alongside its reverse $(B, A)$. This exhaustive pairing strategy allows us to rigorously assess fusion performance across a wide spectrum of semantic relationships—ranging from semantically close concepts to challenging distant combinations, such as fusing a 'violin' with a 'panda' or a 'horse' with 'lipstick'. This also critically highlights our model's ability to generalize and compose novel concepts effectively across diverse domains.

Table 5: List of Objects in the IIOF Dataset by Category.

| Category | Object Names |
|---|---|
| Animals | wolf, panda, owl, rabbit, horse, giraffe, corgi, cat, bird, sheep |
| Fruits | apple, orange, strawberry, durian, lime, pear, pineapple, watermelon, tomato, pepper |
| Artificial Objects | lipstick, violin, coffee cup, rocking horse, glass jar, car, teapot, cake, man, teddy bear |
| Character Figurines | iron man figurine, monkey king figurine, doll figurine, pikachu figurine, charizard figurine, ultraman figurine, astronaut figurine, venusaur figurine, panda figurine, squirtle figurine |

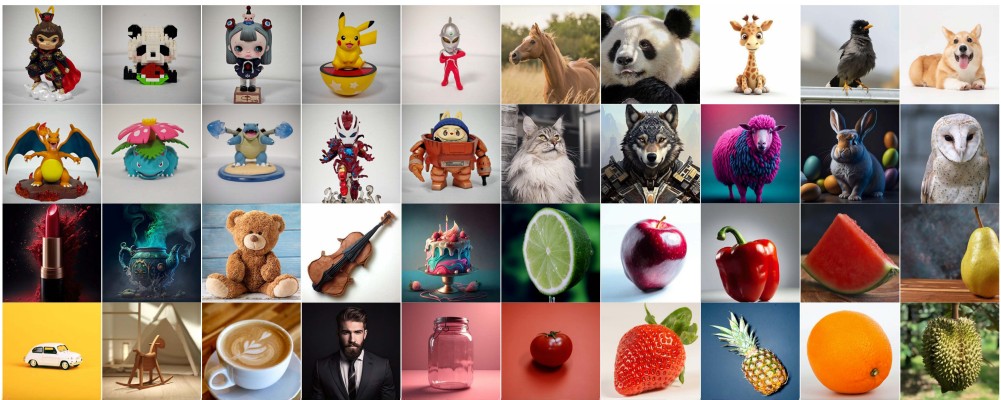

Figure 18: Original Object Image Set.

[4]https://www.pexels.com/

## C  USER STUDY

To evaluate the perceptual quality and human preference for the novel images generated by our fusion framework, we conducted two user studies. These studies assessed our method, **VMDiff**, against state-of-the-art baselines in two main categories: *Multi-Concept Generation* methods and *Mixing and Image Editing* methods. The overall vote distributions are visualized in Fig. 8, while detailed per-example preferences are presented in Table 6 and Table 7. An example user study question for the *Multi-Concept Generation* group and the *Mixing and Image Editing* group is provided in Fig. 19. A total of 76 participants completed the survey, each evaluating 12 fused results (6 from each group), contributing a total of 912 votes. Participants were asked to select the fusion result that best integrated the given concepts in terms of visual quality, creativity, and semantic consistency. As shown in Fig. 8, our method consistently received the highest number of votes in both evaluation groups. In the *Mixing and Image Editing* category (left pie chart), VMDiff garnered a significant **397 votes (87.1%)** of the total. This considerably surpassed other methods such as Stable Flow (Avrahami et al., 2025) (5 votes, 1.1%), ATIH (Xiong et al., 2024) (34 votes, 7.5%), Conceptlab (Richardson et al., 2024) (4 votes, 0.9%) and FreeBlend (Zhou et al., 2025) (16 votes, 3.5%). For instance, as illustrated in Fig. 19, for the "astronaut figurine-monkey king figurine" fusion, our method obtained 81.58% of the votes, demonstrating its strong capability in seamlessly integrating distinct visual elements.

In the *Multi-Concept Generation* category (right pie chart), **VMDiff** led with **307 votes (67.3%)**, significantly outperforming GPT-4o (OpenAI, 2025), which ranked second with 59 votes (12.9%). Other baselines—DreamO (56 votes, 12.3%), MIP-Adapter (17 votes, 3.7%), and OmniGen (17 votes, 3.7%)—received notably fewer votes. In the "doll figurine–corgi" case,

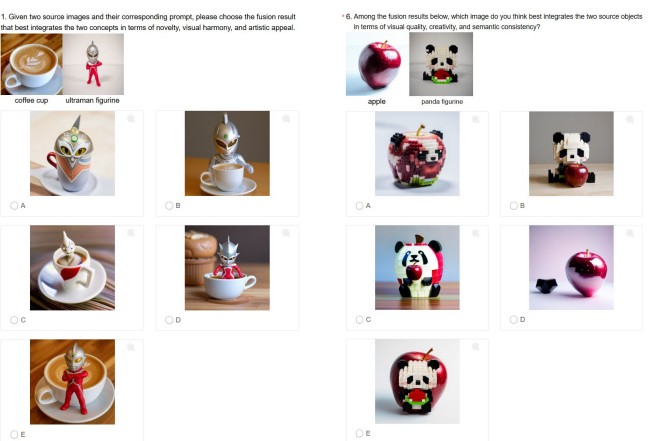

Figure 19: An example of a user study comparing various multi-concept generation, mixing and image editing methods.

VMDiff earned **78.95%** of preferences. Even in more challenging cases like "apple–panda figurine" (see Fig. 19), it maintained an edge with **75.00%** over GPT-4o's **5.26%**. These results indicate that **VMDiff** better aligns with human preferences for visual coherence, creativity, and concept integration, consistently outperforming existing methods across diverse fusion scenarios.

Table 6: User study with multi-concept generation methods.

| image-image | A(Our VMDiff) | B(DreamO) | C(MIP-Adapter) | D( OmniGen) | E(GPT-4o) |
|---|---|---|---|---|---|
| coffee cup-ultraman figurine | 43(56.58%) | 11(14.47%) | 7(9.21%) | 3(3.95%) | 12(15.79%) |
| sheep-car | 57(75.00%) | 4(5.26%) | 1(1.32%) | 2(2.63%) | 12(15.79%) |
| doll figurine-corgi | 60(78.95%) | 1(1.32%) | 3(3.95%) | 3(3.95%) | 9(11.84%) |
| lime-glass jar | 45(59.21%) | 22(28.95%) | 1(1.32%) | 0(0.00%) | 8(10.53%) |
| cake-owl | 45(59.21%) | 5(6.58%) | 3(3.95%) | 9(11.84%) | 14(18.42%) |
| apple-panda figurine | 57(75.00%) | 13(17.11%) | 2(2.63%) | 0(0.00%) | 4(5.26%) |

Table 7: User study with mixing and image editing methods.

| image-image | A(Our VMDiff) | B(Stable Flow) | C(ATIH) | D(Conceptlab) | E(FreeBlend) |
|---|---|---|---|---|---|
| astronaut figurine-monkey king figurine | 62(81.58%) | 2(2.63%) | 7(9.21%) | 1(1.32%) | 4(5.26%) |
| man-pikachu figurine | 68(89.47%) | 0(0.00%) | 4(5.26%) | 1(1.32%) | 3(3.95%) |
| doll figurine-panda | 62(81.58%) | 0(0.00%) | 13(17.11%) | 1(1.32%) | 0(0.00%) |
| iron man figurine-charizard figurine | 69(90.79%) | 3(3.95%) | 3(3.95%) | 0(0.00%) | 1(1.32%) |
| squirtle-wolf | 66(86.84%) | 0(0.00%) | 4(5.26%) | 1(1.32%) | 5(6.58%) |
| ultraman figurine-venusaur figurine | 70(92.11%) | 0(0.00%) | 3(3.95%) | 0(0.00%) | 3(3.95%) |

## D    Limitations

Our method effectively fuses two input images into a coherent hybrid object that captures broad conceptual information; however, it has two main limitations. First, inference relies on iterative optimization, which increases computational cost and latency (Table 8). A promising remedy is to train a lightweight prediction/refinement module that guides the fusion in a single forward pass, thereby reducing runtime while maintaining—or even improving—visual quality and semantic balance. Second, in a small fraction of cases the fused outputs do not fully align with human preferences (Fig. 20), exhibiting semantic

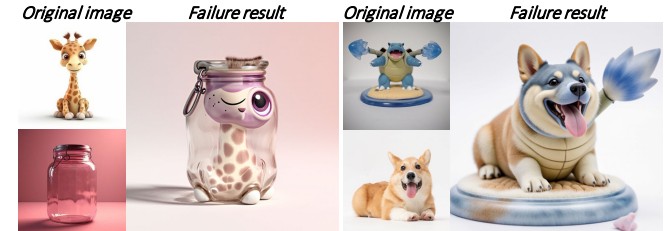

Figure 20: Examples of failure cases where our method produces fused outputs with suboptimal semantic or stylistic coherence.

inconsistencies or stylistic imbalance. Although repeated noise resampling and selection can mitigate these failures, this heuristic has limited controllability. In future work, we will pursue more controllable, preference-aligned fusion via explicit human feedback, aesthetic priors, or learned alignment objectives, enabling results that more reliably reflect human intent and aesthetics.

Table 8: Runtime comparison across methods.

| Methods | Avg. Time / Pair |
|---|---|
| Ours | 2 min 46 sec |
| ATIH | 10 sec |
| Stable Flow | 27 sec |
| Conceptlab | 13 min 45 sec |
| FreeCustom | 22 sec |
| OmniGen | 53 sec |
| Freeblend | 12 sec |
| MIP-Adapter | 12 sec |
| DreamO | 8 sec |

## E    Statement on LLM Usage

In accordance with the ICLR policy on the use of Large Language Models (LLMs), we hereby declare that an LLM (ChatGPT, GPT-5) was used solely to aid or polish the writing of this paper, such as improving grammar and wording. All ideas, technical content, and experimental results are entirely our own. Further details are described within the paper. The authors take full responsibility for the accuracy and integrity of the content.

## F    Algorithm

Algorithm 1 outlines the complete inference process of our proposed framework, **VMDiff**, which integrates a noise refinement step and an efficient adaptive adjustment (EAA) loop. Given two input images $I_1, I_2$ and their category labels $T_1, T_2$, we construct a prompt $P_G$ and initialize the fusion parameters $\theta = \{\alpha, \beta_1, \beta_2, \epsilon\}$.

The algorithm begins by sampling initial Gaussian noise $\epsilon$, which is refined through a denoising-inversion procedure to produce a structure-aware latent representation $\epsilon_r$. The core loop involves:

- **Searching** for the optimal interpolation factor $\alpha$ using Golden Section Search to maximize the similarity score $S(\theta)$.
- **Conditionally adjusting** the noise scaling factors $\beta_1, \beta_2$ when the current fusion score is below a threshold $TH$, guiding the fusion toward balance between the two source objects.

---

**Algorithm 1:** VMDiff with Efficient Adaptive Adjustment (VMDiff-EAA)

---

**Input:** images $I_1, I_2$, labels $T_1, T_2$, prompt $P_G$, threshold $TH$, max rounds $K$
**Output:** fused image $I^*$ and parameters $\theta^* = \{\alpha^*, \beta_1^*, \beta_2^*, \epsilon^*\}$

1  Compute embeddings $z_1 = \mathcal{E}_I(I_1)$, $z_2 = \mathcal{E}_I(I_2)$, $z_p = \mathcal{E}_T(P_G)$;
2  Initialize $\alpha = 0.5$, $\beta_1 = \beta_2 = 1.0$;  $S_{\text{best}} = -\infty$, $\theta_{\text{best}} = \varnothing$;
3  **for** $k = 1$ **to** $K$ **do**
4  $\quad$ Sample noise $\epsilon \sim \mathcal{N}(0, I)$;
5  $\quad$ $z_{\text{SCat}} = \text{concat}(\beta_1 z_1, \beta_2 z_2)$, $x_T = \epsilon$;
6  $\quad$ **for** $t = T$ **to** $t_{den}$ **do**
7  $\quad\quad$ $x_{t-1} = x_t - (\sigma_t - \sigma_{t-1})v_\phi(x_t, t, z_{\text{SCat}}, \gamma_{\text{den}}, z_p)$
8  $\quad$ **for** $t = t_{den}$ **to** $T$ **do**
9  $\quad\quad$ $x_{t+1} = \hat{x}_t + (\sigma_{t+1} - \sigma_t)v_\phi(\hat{x}_t, t, z_{\text{SCat}}, \gamma_{\text{inv}}, z_p)$
10  $\quad$ $\epsilon_r = \hat{x}_T$;
11  $\quad$ $\alpha^* = \text{GoldenSearch}(\alpha \in [0, 1], f(\alpha) = S(\alpha, \beta_1, \beta_2, \epsilon_r))$;
12  $\quad$ $(S, S_{I_1}, S_{I_2}, S_{T_1}, S_{T_2}) = \text{Score}(\alpha^*, \beta_1, \beta_2, \epsilon_r)$;
13  $\quad$ **if** $S > S_{best}$ **then**
14  $\quad\quad$ $S_{\text{best}} = S$; $\theta_{\text{best}} = \{\alpha^*, \beta_1, \beta_2, \epsilon_r\}$
15  $\quad$ **if** $S \geq TH$ **then**
16  $\quad\quad$ **return** $I(\theta^*)$, $\theta^*$
17  $\quad$ $S_1 = S_{I_1} + S_{T_1}$, $S_2 = S_{I_2} + S_{T_2}$;
18  $\quad$ **if** $S_1 > S_2$ **then**
19  $\quad\quad$ $\beta_2^* = \text{GoldenSearch}(\beta_2 \in [\beta_{\min}, \beta_{\max}], f(\beta_2))$
20  $\quad$ **else**
21  $\quad\quad$ $\beta_1^* = \text{GoldenSearch}(\beta_1 \in [\beta_{\min}, \beta_{\max}], f(\beta_1))$
22  $\quad$ $(S', \cdot) = \text{Score}(\alpha^*, \beta_1^*, \beta_2^*, \epsilon_r)$;
23  $\quad$ **if** $S' > S_{best}$ **then**
24  $\quad\quad$ $S_{\text{best}} = S'$; $\theta_{\text{best}} = \{\alpha^*, \beta_1^*, \beta_2^*, \epsilon_r\}$
25  $\quad$ **if** $S' \geq TH$ **then**
26  $\quad\quad$ Normalize $z_1, z_2$ and compute spherical interpolation $z_{\text{SInp}}(\alpha^*)$;
27  $\quad\quad$ $x_T = \epsilon_r$;
28  $\quad\quad$ **for** $t = T$ **to** $0$ **do**
29  $\quad\quad\quad$ $x_{t-1} = x_t - (\sigma_t - \sigma_{t-1})v_\phi(x_t, t, z_{\text{SInp}}(\alpha^*), \gamma_{\text{gen}}, z_p)$
30  $\quad\quad$ $I = \mathcal{D}(x_0)$;  **return** $I$, $\theta^*$
31  **if** $\theta_{best} \neq \varnothing$ **then**
32  $\quad$ Decode best parameters $\theta_{\text{best}}$ via MixingDenoise;
33  $\quad$ **return** $I$, $\theta_{\text{best}}$
34  **return** $\varnothing$;

---

- **Returning** a fused image $I(\theta^*)$ once a satisfactory similarity score is achieved.

This design ensures a lightweight and interpretable optimization routine over a low-dimensional parameter space. The algorithm reliably produces perceptually and semantically coherent hybrid images, as validated in our experiments.

## G  MORE RESULTS

In this section, we present additional qualitative results with **resampling disabled**, to evaluate VMDiff under a deterministic setting and further demonstrate its effectiveness and generalization. Fig. 1 shows generations at $1024 \times 1024$ resolution. Figs. 21, 22, 23, 24, 25, 26, 27, 28, and 29 provide diverse fusion examples spanning animals, fruits, artificial objects, and character figurines. In all figures, the leftmost column displays the source images, and the adjacent columns show the fused outputs.

These examples are generated from our IIOF dataset and cover a wide range of visual appearances and semantic attributes. Across varied fusion types—such as person–fruit, animal–object, and object–object—the results consistently exhibit structural coherence, balanced integration, and high visual fidelity. This indicates that VMDiff can integrate symbolic and structural cues into stylistically consistent hybrids, regardless of whether the source concepts are semantically similar or dissimilar.

Overall, these results substantiate the strong generalization of VMDiff, yielding novel, imaginative, and structurally plausible hybrid objects from diverse real-world inputs, even without resampling or seed variation.

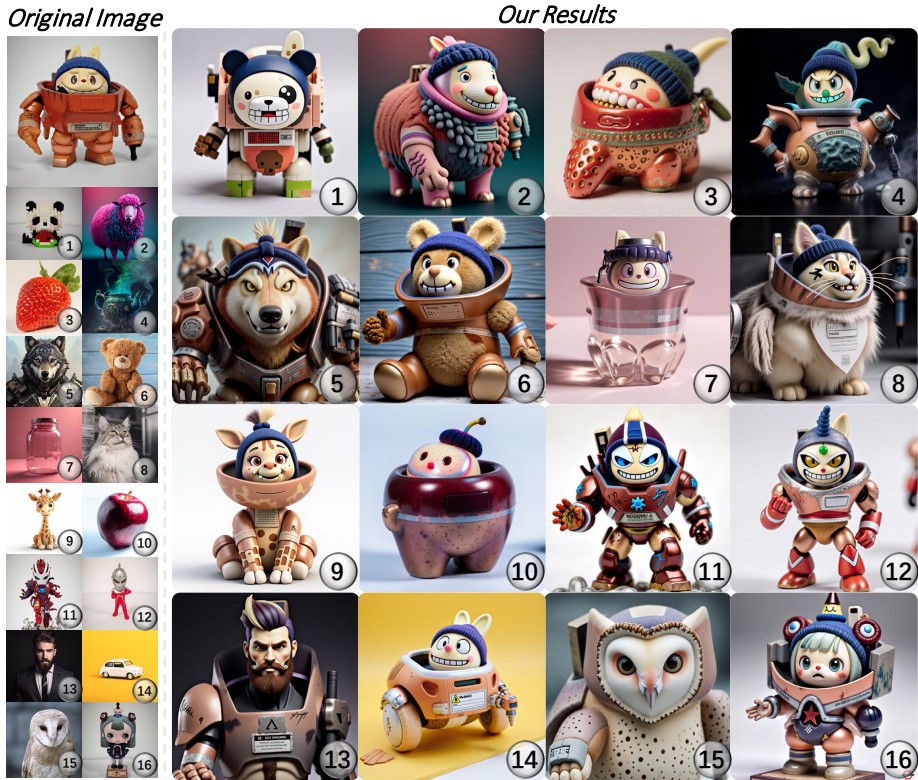

Figure 21: **More Results.** The primary source (*astronaut figurine*, top-left) is fused with secondary inputs (left column), with results shown on the right.

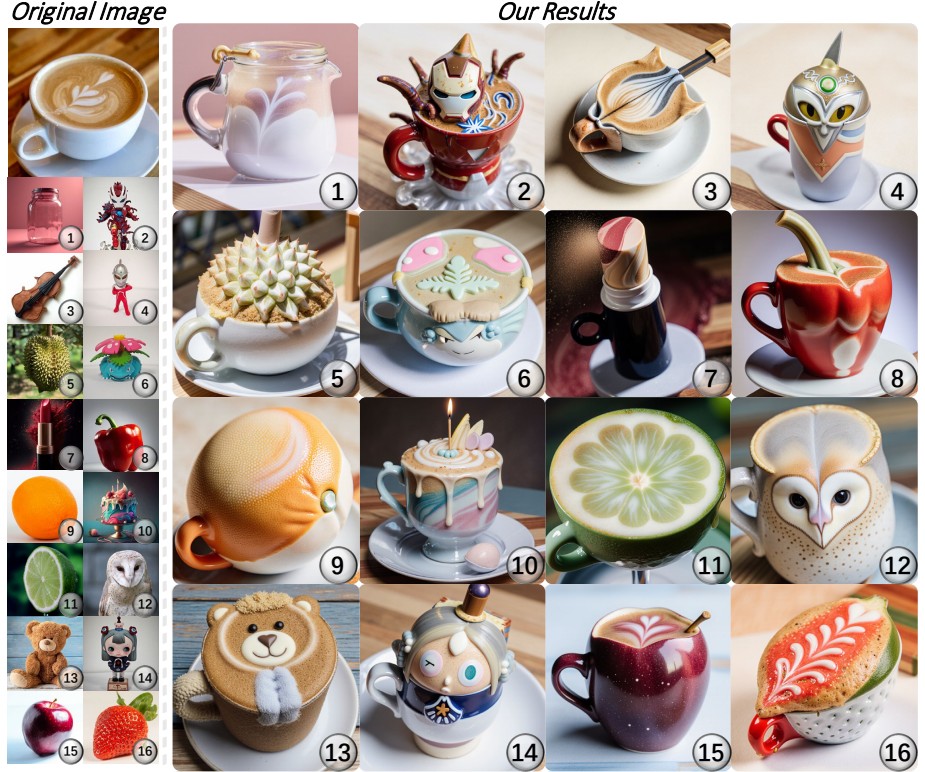

Figure 22: **More Results.** The primary source (*coffee cup*, top-left) is fused with secondary inputs (left column), with results shown on the right.

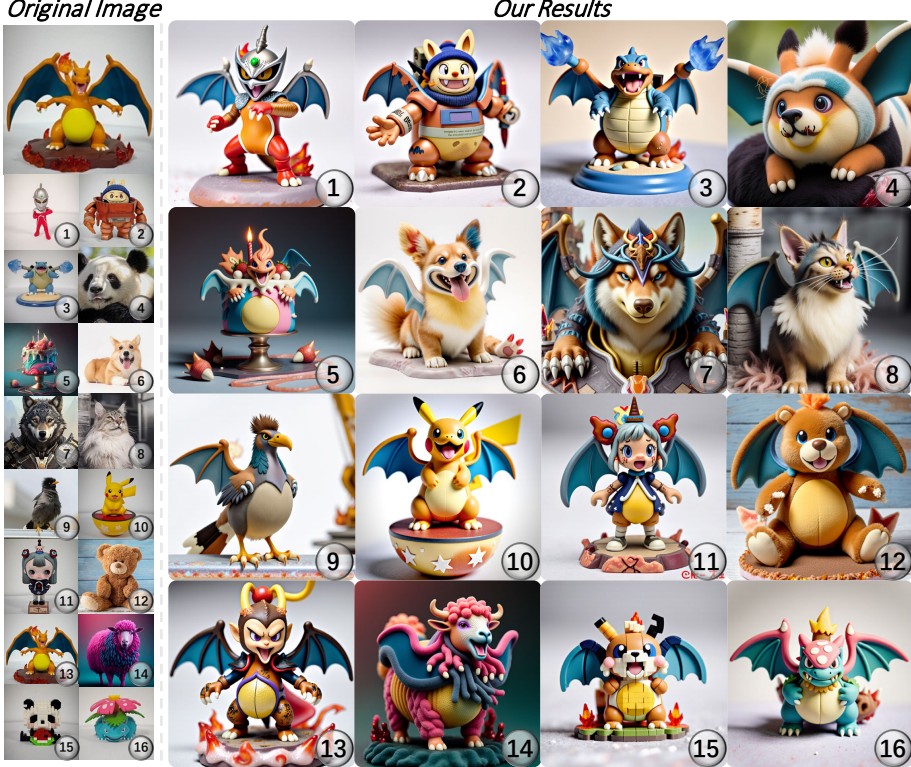

Figure 23: **More Results.** The primary source (*charizard figurine*, top-left) is fused with secondary inputs (left column), with results shown on the right.

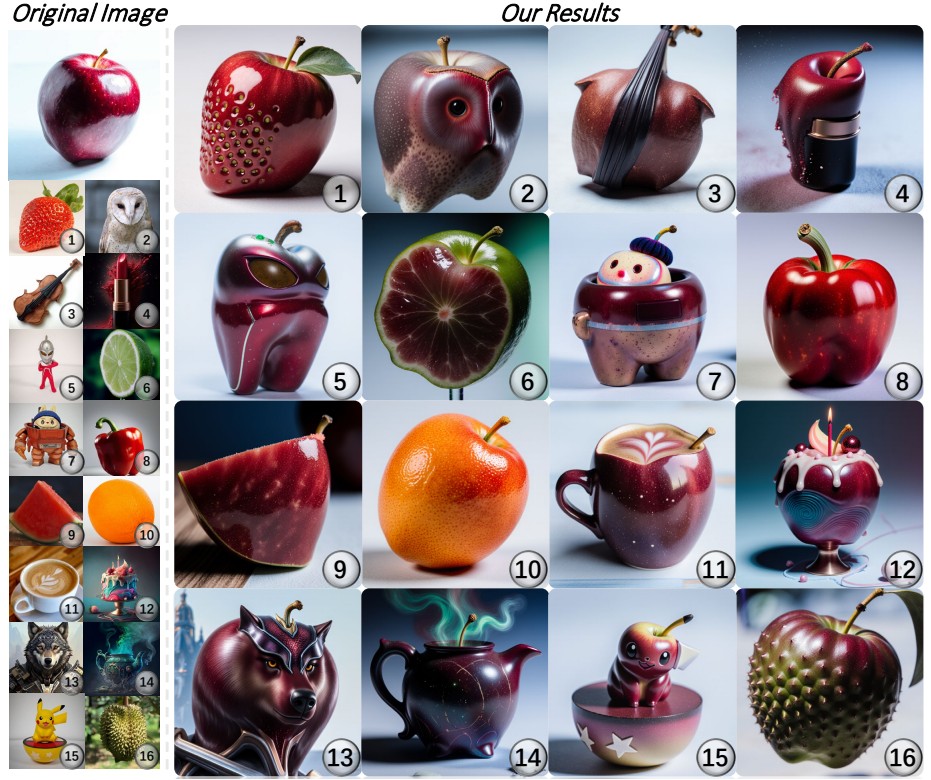

Figure 24: **More Results.** The primary source (*apple*, top-left) is fused with secondary inputs (left column), with results shown on the right.

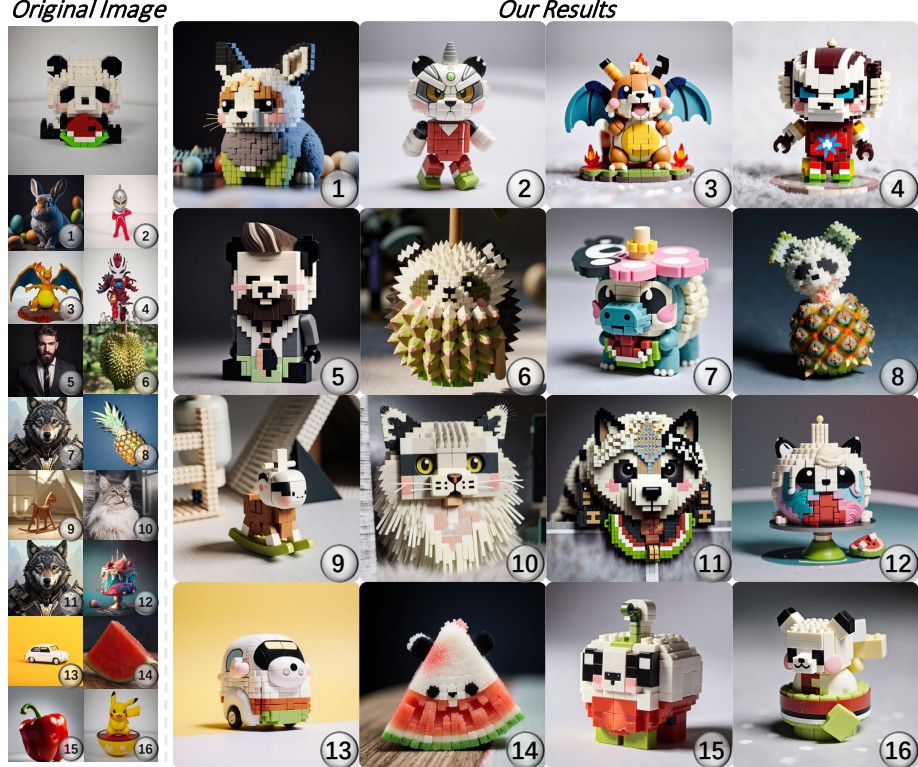

Figure 25: **More Results.** The primary source (*panda figurine*, top-left) is fused with secondary inputs (left column), with results shown on the right.

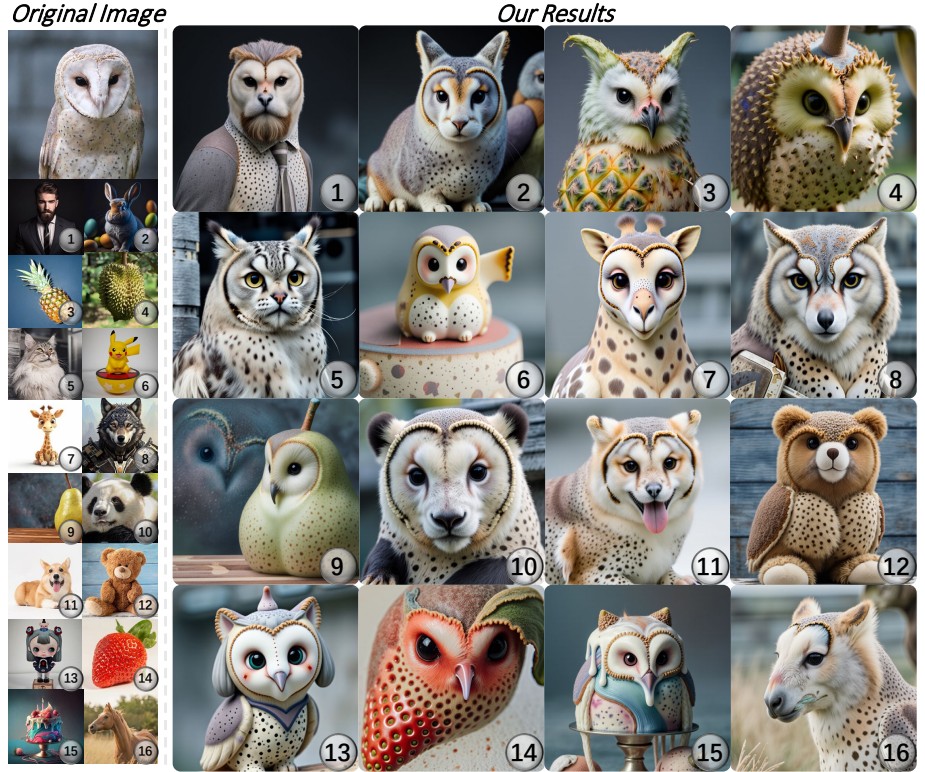

Figure 26: **More Results.** The primary source (*owl*, top-left) is fused with secondary inputs (left column), with results shown on the right.

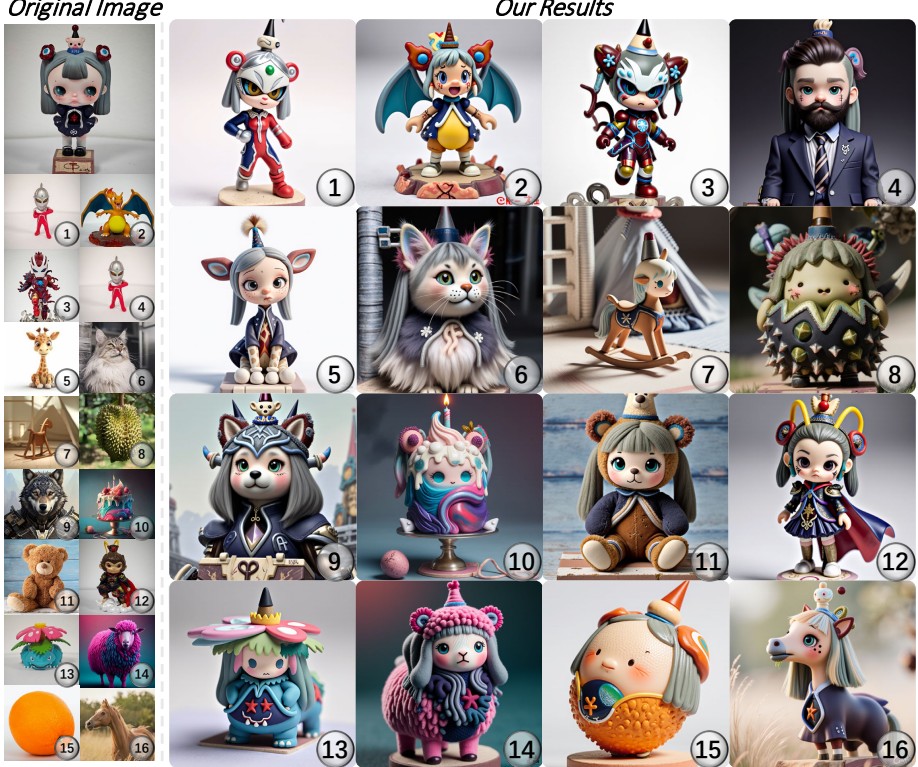

Figure 27: **More Results.** The primary source (*doll figurine*, top-left) is fused with secondary inputs (left column), with results shown on the right.

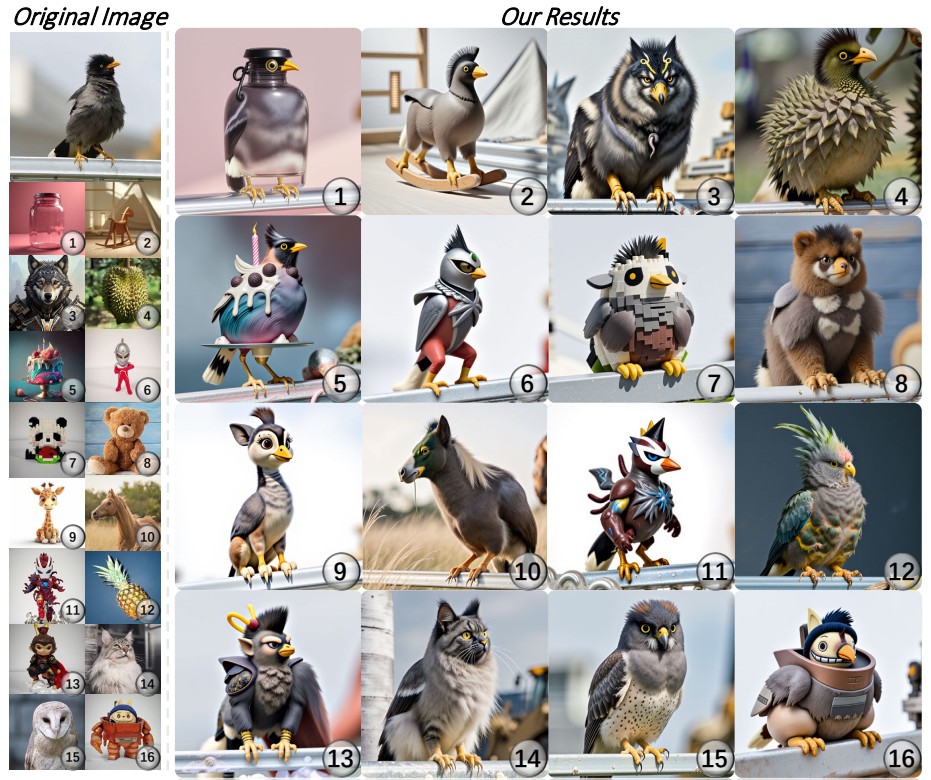

Figure 28: **More Results.** The primary source (*bird*, top-left) is fused with secondary inputs (left column), with results shown on the right.

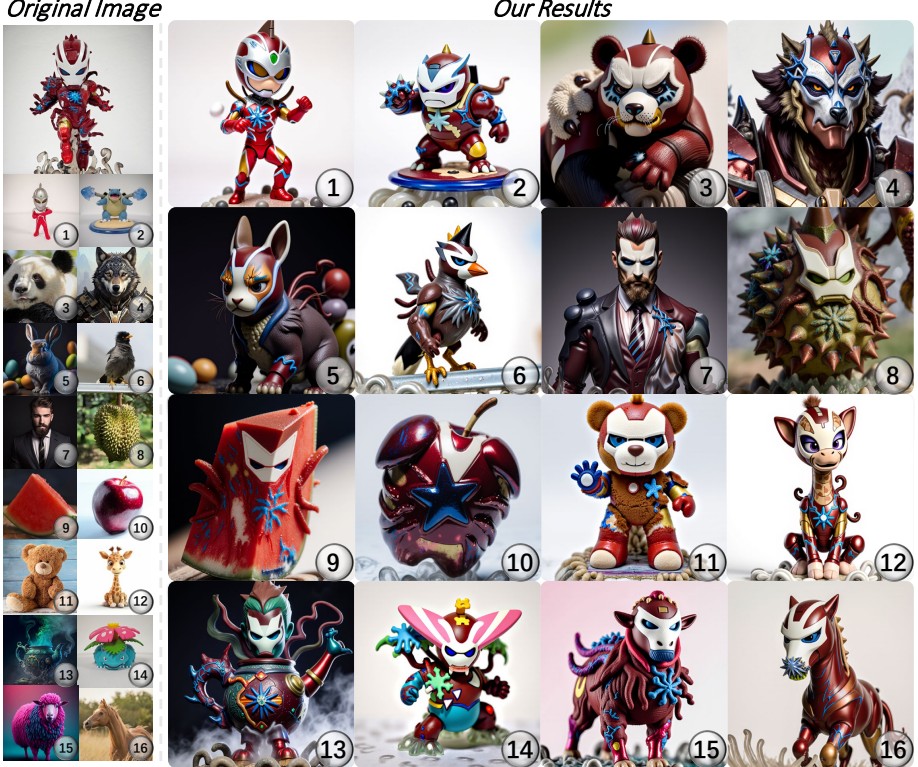

Figure 29: **More Results.** The primary source (*Iron man figurine*, top-left) is fused with secondary inputs (left column), with results shown on the right.