# OpenReview forum: "VMDiff: Visual Mixing Diffusion for Limitless Cross-Object Synthesis"
_ICLR.cc/2026/Conference — ICLR 2026 Poster_

### Official Review · Reviewer_4syL · 2025-10-28

**Soundness:** 3
**Presentation:** 2
**Contribution:** 3
**Rating:** 6
**Confidence:** 2

**Summary:**

This paper introduces an inference-time method for generating novel, visually coherent hybrid objects from customized image inputs. It requires no retraining or fine-tuning of the underlying diffusion model; instead, it optimizes a small set of fusion parameters during inference to effectively combine visual and semantic information from two distinct sources.

The authors identify two key failure modes in existing pretrained models when performing cross-object synthesis: coexistent generation, where objects are merely overlaid or placed side-by-side without true integration, and bias generation, where one object dominates the output while the other is largely ignored due to semantic imbalance. To address these issues, VMDiff proposes a Hybrid Sampling Process (HSP) and an Efficient Adaptive Adjustment (EAA) module. HSP integrates inputs at both the noise and latent levels, enabling seamless fusion through curvature-aware spherical interpolation. Meanwhile, EAA automatically searches for optimal fusion parameters by maximizing a similarity-based score that balances visual fidelity, semantic consistency, and inter-object symmetry.

Experiments on the newly curated IIOF benchmark (780 images) demonstrate that VMDiff consistently outperforms state-of-the-art baselines. User studies also show a strong preference for VMDiff’s outputs.

**Strengths:**

1. This work clearly defines the visual mixing task and identifies two key failure modes—coexistent and bias generation—in existing methods.
2. The proposed Hybrid Sampling Process effectively combines noise-level and latent-level fusion, enabling both detail preservation and semantic integration.
3. The method is evaluated on a new benchmark with diverse object pairs and shows consistent improvements over strong baselines in both automatic metrics and human studies.
4. This work shows a clear generation process of the proposed pipeline

**Weaknesses:**

1. **Lack of multi-image generalization**: All experiments are limited to pairwise fusion; the method’s extensibility to three or more input images is neither explored nor validated.

2. **No cross-model evaluation**: As an inference-time approach, its compatibility with other diffusion backbones (e.g., SD3, SDXL) is not tested, leaving the potential of this method unclear

3. **Missing time-cost ablation**: Owing to high inference latency (166 sec/pair), the paper should break down the computational overhead of individual components (e.g., BNoise, α/β search, SS scoring), to justify the necessity of each design

**Questions:**

1. Is manual intervention feasible—for instance, to selectively enhance the identity features of one image or the stylistic attributes of another?
2. In certain scenarios, fine-grained control over specific regions may be required—such as replacing a human hand with a robotic arm, or even transforming a foot into a robotic arm. Can this method reliably discern subtle distinctions in the prompt and generate outputs that accurately reflect such targeted modifications?

---

> ### Author Response · Authors · 2025-11-21
>
> We thank reviewer 4syL for the positive assessment of our task formulation, HSP+EAA design, and experimental results on IIOF. In the revised PDF, we include additional experiments on multi-image fusion, cross-model evaluation, and a more detailed runtime breakdown, together with new figures illustrating these aspects. Below, we address the concerns about multi-image generalization, compatibility with other backbones, time-cost analysis, and the possibilities and limitations of manual and fine-grained control.
>
> **Q1: On multi-image generalization.**
>
> **A1:** We agree that our main experiments focus on pairwise fusion. To probe extensibility beyond two inputs, we conducted preliminary trials with three images, as illustrated in Fig 11. We use a simple sequential strategy: first apply VMDiff to $(I_1,I_2)$, then fuse the resulting hybrid with $I_3$. This straightforward extension can still produce visually coherent three-way hybrids that blend attributes from all three sources, indicating that our formulation can be applied beyond pairwise fusion. However, compared with the two-image setting, we observe more pronounced information loss and imbalance across the three contributors. In light of these limitations, we deliberately scope this work to image pairs and regard robust multi-object fusion as an important direction for future research. One promising avenue is to design a lightweight, permutation-invariant module that aggregates a variable number of image embeddings into a single fused embedding, which can then be plugged into our framework to better support multi-image fusion.
>
> ---
>
> **Q2: Backbone generalization.**
>
> **A2:** This is an important point and we agree that the backbone should be decoupled from the proposed fusion mechanism. Conceptually, HSP+EAA is backbone-agnostic: it only assumes access to image-conditioned embeddings that can be mixed, and does not depend on any Flux-specific architecture. In practice, however, the $interface$ through which image information is injected is crucial for how much instance-level detail our fusion can preserve.
>
> Fig 12 applies our framework to three backbones: Flux-1.0-dev with the Redux image encoder, SDXL+IP-Adapter, and SD-3.5+IP-Adapter. With the same image pairs, fusion prompt, and hyperparameters, all backbones can often generate single hybrid objects, indicating that our method is not tailored to Flux-Krea alone. Flux+Redux encodes images into the same semantic latent space as text, so our BNoise+SInp mixing directly operates on rich, text-like image embeddings and best retains object-specific geometry and appearance from both sources. For SDXL and SD-3.5, image information is provided via IP-Adapter [ref-1], which keeps the text prompt fixed and injects image features as additional attention tokens; interpolating these tokens mainly modulates coarse, prompt-level semantics and leads to more loss of fine instance details.
>
> Overall, these results suggest that VMDiff is most effective on backbones that (i) preserve rich instance-level information from the inputs, and (ii) expose this information as compact image embeddings that live in a semantic space compatible with text (as Redux does). Our SDXL/SD-3.5 experiments should therefore be viewed as feasibility checks under a weaker image interface rather than as the ideal deployment setting. We expect that equipping SDXL, SD-3.5, or other flow models with a Redux-style semantic image encoder could substantially narrow the fusion-quality gap to Flux-Krea, and may enable performance comparable to our Flux-based results.
>
> [ref-1] Ye H, Zhang J, Liu S, et al. Ip-adapter: Text compatible image prompt adapter for text-to-image diffusion models. arXiv, 2023.
>
> ---

---

> ### Author Response · Authors · 2025-11-21
>
> **Q3: Time-cost ablation.**
>
> **A3:** We have now added a component-wise runtime breakdown measured on NVIDIA RTX 4090. In our longest runs for one fused pair under the full HSP+EAA setting, the costs are:
>
> - **BNoise.** The conditional denoise--invert cycle used to construct blended noise takes about 3\,s per pair. As shown in Fig 14 and Table 4, this semantic initialization helps preserve more structure from both inputs than pure Gaussian noise, so we keep it.
>
> - **$\alpha/\beta$ search.** The dominant cost comes from repeated denoising trajectories with different $(\alpha,\beta_1,\beta_2)$. In the most expensive cases, the cumulative time spent on the $\alpha$-search stage can reach up to 95.57s, and the $\beta$-search stage up to 93.33s, since each candidate requires a (partial) forward denoising pass. This search is what replaces manual tuning and allows us to automatically find balanced fusions across diverse input pairs.
>
> - **SS scoring.** Computing the similarity score $S(\theta)$ for each candidate (image/text CLIP/LMM forwards) takes below 1s and is negligible compared to the denoising cost, while providing the objective that guides the $\alpha/\beta$ search.
>
> In practice, the actual wall-clock time is often smaller than this worst-case configuration, because EAA supports early stopping: once an $\alpha$ configuration achieves a score $S(\theta)$ above the acceptance threshold, we skip $\beta$-search. For interactive use, users may also disable $\beta$-search or coarsen the $\alpha/\beta$ grid to trade a small loss in optimality for a substantial speedup.
>
> ---
>
> **Q4: Manual intervention for identity/style bias.**
>
> **A4:** Our framework does support manual intervention at the *global* level. Since VMDiff optimizes a similarity-based fusion score $S(\theta)$ over image/text embeddings, users can steer the fusion toward one source by adjusting the relative weights of the similarity terms linked to $I_1$ vs. $I_2$, or by biasing the mixing parameters (e.g., $\alpha$ or the text/image weights) so that one image contributes more strongly to the fused embedding. In practice, this allows users to emphasize identity features from one source or global stylistic traits from the other, while still generating a *single* hybrid object rather than two coexisting ones. However, such control remains coarse: it affects the overall object instead of isolating individual parts.
>
> ---
>
> **Q5: Fine-grained control over local edits.**
>
> **A5:** Our current design focuses on *global* hybrid-object synthesis rather than fine-grained image editing. VMDiff fuses two inputs by mixing their image/text embeddings and optimizing a global fusion score $S(\theta)$, without explicit part-level masks or region-wise conditioning. As a result, it is not intended to reliably support highly localized edits such as only replace the human hand by a robotic arm or turn the foot into a robotic arm, and we agree that subtle, part-specific prompt distinctions are not guaranteed to be faithfully realized.
>
> Extending VMDiff to true region-level or attribute-specific control (e.g., hand-only, style-only) would require augmenting our framework with explicit spatial constraints or part-aware conditioning modules, and possibly integrating it with controllable diffusion techniques. We view this as an important direction for future work, and complementary to our current focus on globally coherent hybrid objects.

---

> > ### Comment · Reviewer_4syL · 2025-11-24
> >
> > Thank you for your response. In light of the new content and results you have provided addressing my concerns, I am now convinced that your method is both effective and generalizable. Therefore, I believe this paper clearly meets the acceptance criteria. I will maintain my positive rating and increase my confidence score accordingly.

---

> > > ### Author Response · Authors · 2025-11-24
> > >
> > > We sincerely thank the reviewer for the warm follow-up and for taking the time to re-evaluate our revisions. We’re very glad that the new analyses and experiments addressed your concerns about effectiveness and generalizability. We truly appreciate your decision to keep the positive rating and increase the confidence score.

---

### Official Review · Reviewer_wsRz · 2025-10-29

**Soundness:** 2
**Presentation:** 3
**Contribution:** 2
**Rating:** 4
**Confidence:** 4

**Summary:**

This paper proposes a framework (VMDiff) for visual cues fusion and mitigates the coexistent generation and bias generation problems that exist in previous works. The pipeline incorporates two important modules: the hybrid sampling process and the adaptive parameter adjustment for efficient latent space manipulation for semantic information integration and conditioning. VMDiff generally outperforms previous approaches under different evaluation settings.

**Strengths:**

1. The method proposed in the paper introduces free lunch, which can be seamlessly integrated into different diffusion architectures, which could be an interesting direction to explore the general usage cases.

2. Detailed information on design choices over qualitative figures and quantitative numbers is given to showcase the effectiveness of the proposed approaches.

3. A benchmark is proposed to demonstrate the effectiveness of the proposed method.

**Weaknesses:**

1. The quantitative evaluation details are ambiguous. For example, in Table 1, apart from the proposed approach, MIP-Adapter outperforms OmniGen and DreamO; however, this seems opposite in Figure 6, where MIP-Adapter is less photorealistic compared to all others, which can also be demonstrated by Table 5 in the user study. This phenomenon makes the values less convincing.

2. In the ablation study, most value improvements are introduced by MDNoise; the changes brought by BNoise are not obvious.

3. The paper proposes a method focusing on the fusion of two different objects into another one. In some baselines, such as DreamO and OmniGen, they target a broader and more generalizable application (supporting different prompts with the coexistence of two objects). It would be more intuitive to compare with other papers targeting the same application scale. Besides, it seems this framework is training-free. It could help to compare with other training-free approaches, which might help increase the soundness of the proposed method.

**Questions:**

1. Evaluation: (a). It would help to validate the comparison if more elaborations or explanations about the evaluation could be provided based on the questions in weakness 1. (b) It's interesting to see how different values given under these metrics for different inputs are semantically good. What are the prompts? Are they all the same? For example, the first column in Figure 6, the results about VMDiff, DreamO, OmniGen, and GPT-4o all seem valid.
2. In the Table 2 ablation part, the comparisons between rows 1 vs. 2 and 3 vs. 4 indicate that the improvements from pure BNoise and $\beta$ search are not obvious. It would help if we had more scores in the setting baseline1 vs baseline 2 vs baseline 1+ BNoise ($\beta$ search) to isolate the improvement caused by BNoise; setting: baseline 1 vs. random noise + MDNoise ($\alpha$ search) to isolate the influence of  MDNoise. It is currently not obvious why the $\epsilon_b$ incorporates more semantic information instead of just Gaussian noise differently sampled.
3. Is this framework commutative? For example, will "a photo of <$T_1$> fused with <$T_2$>" be more similar to $I_1$ while "a photo of <$T_2$> fused with <$T_1$>" is inclined to generate an object similar to $I_2$?
4. How to isolate the improvement proposed by base models? Specifically, the MIP-Adapter is based on sdxl, DreamO is under Flux-1.0-dev, with Krea is more updated. Would some quick validation on these base models using the proposed framework help prove the effectiveness?
5. One possible problem with the current evaluation of concatenation in the second row is that the $\alpha$ changes so small, which may influence a lot under the interpolated space but not be obvious when two latens are separately used. What would happen for other simple values, such as concat($\alpha z_1,(1-\alpha)z_2$)

---

> ### Author Response · Authors · 2025-11-21
>
> We thank reviewer wsRz for the constructive feedback and for recognizing the interest of a training-free, plug-and-play fusion framework with detailed analyses and a dedicated benchmark. We have expanded the quantitative evaluation and added new ablations and qualitative figures in the revised PDF. Below, we clarify our metrics and prompts, disentangle BNoise vs. MDeNoise, discuss baseline scope and training-free comparisons, and answer the questions on commutativity, backbone dependence, and the concatenation baseline.
>
> **Q1: Metric scores vs. visual realism inconsistency.**
>
> **A1:** We appreciate the reviewer’s careful cross-checking of the tables and qualitative examples. The key point is that Table 1 is designed to evaluate *semantic fusion quality*, not photo-realism. Specifically, our SA/SCE metrics are LMM-based measures that focus on (i) semantic alignment with the fusion prompt and (ii) single-entity consistency, rather than overall realism or aesthetics.
>
> As shown by additional qualitative comparisons in Fig 17, MIP-Adapter tends to keep both source text categories within a single object structure, which matches the reward pattern of SA/SCE: it often generates one fused creature that contains recognizable cues from both textual categories and aligns well with the fusion prompt, even though many fine-grained appearance details from the original inputs are discarded. In contrast, OmniGen and DreamO frequently produce visually impressive, highly realistic images, but they often (i) omit one of the two source concepts or (ii) generate two coexisting objects instead of a single hybrid; such behaviours are explicitly penalized by our SA/SCE/SS metrics.
>
> This explains why MIP-Adapter ranks higher than OmniGen/DreamO in Table 1, yet appears less realistic in Fig 6 and receives lower user preference in Table 5, where human raters are more sensitive to realism and overall visual appeal.
>
> ---
> **Q2: Clarification of evaluation metrics and prompts.**
>
>
> **A2:** We thank the reviewer for the question about prompts. For all methods in Fig 6 (including VMDiff, DreamO, OmniGen, MIP-Adapter, and GPT-4o), we use the *same* inputs: the same pair of source images $(I_1,I_2)$ and a single fixed text prompt
> "A photo of $T_1$ creatively fused with $T_2$."
> Here $T_1$ and $T_2$ are the category names. This shared prompt is used both for generation and for our evaluation.
> Our metrics further distinguish how well each result (i) preserves both source categories within a single entity and (ii) avoids collapsing to one side or producing two separate objects, which is why their scores still differ.
>
> ---
>
>
> **Q3:** Is the framework commutative w.r.t. text order?
>
> **A3:**  Our framework is not strictly commutative in the mathematical sense, because the text encoder can respond slightly differently to  "A photo of $T_1$ creatively fused with $T_2$." and   "A photo of $T_2$ creatively fused with $T_1$."
>
> To isolate this effect, we first examine the ``Random noise + MDeNoise ($\alpha=0.5$)'' baseline, where no adaptive search is involved. We keep the image pair $(I_1,I_2)$ and all other settings fixed, and only swap the order of $T_1$ and $T_2$ in the prompt. As shown in Fig 15, for most pairs (e.g., *panda figurine + cat* or *pepper + lime*), the two orderings produce very similar hybrids. In a few harder cases (right-most example), the leading token receives slightly more emphasis and a few additional attributes may appear, but in both orderings the result remains a single coherent hybrid rather than collapsing to one source.
>
> In the full VMDiff pipeline, this mild asymmetry is further reduced by our symmetric fusion score $S(\theta)$ and the EAA search: both image sources enter the score in a balanced way, and candidates that overly favor one category are explicitly penalized. As a result, while the framework is not mathematically commutative in text order, in our experiments the two orderings most often yield similar hybrids, and any preference toward $T_1$ or $T_2$ appears mild and case-dependent rather than systematic.
>
> ---

---

> ### Author Response · Authors · 2025-11-21
>
> **Q4: Backbone generalization.**
>
> **A4:** This is an important point and we agree that the backbone should be decoupled from the proposed fusion mechanism. Conceptually, HSP+EAA is backbone-agnostic: it only assumes access to image-conditioned embeddings that can be mixed, and does not depend on any Flux-specific architecture. In practice, however, the $interface$ through which image information is injected is crucial for how much instance-level detail our fusion can preserve.
>
> Fig 12 applies our framework to three backbones: Flux-1.0-dev with the Redux image encoder, SDXL+IP-Adapter, and SD-3.5+IP-Adapter. With the same image pairs, fusion prompt, and hyperparameters, all backbones can often generate single hybrid objects, indicating that our method is not tailored to Flux-Krea alone. Flux+Redux encodes images into the same semantic latent space as text, so our BNoise+SInp mixing directly operates on rich, text-like image embeddings and best retains object-specific geometry and appearance from both sources. For SDXL and SD-3.5, image information is provided via IP-Adapter [ref-1], which keeps the text prompt fixed and injects image features as additional attention tokens; interpolating these tokens mainly modulates coarse, prompt-level semantics and leads to more loss of fine instance details.
>
> Overall, these results suggest that VMDiff is most effective on backbones that (i) preserve rich instance-level information from the inputs, and (ii) expose this information as compact image embeddings that live in a semantic space compatible with text (as Redux does). Our SDXL/SD-3.5 experiments should therefore be viewed as feasibility checks under a weaker image interface rather than as the ideal deployment setting. We expect that equipping SDXL, SD-3.5, or other flow models with a Redux-style semantic image encoder could substantially narrow the fusion-quality gap to Flux-Krea, and may enable performance comparable to our Flux-based results.
>
> [ref-1] Ye H, Zhang J, Liu S, et al. Ip-adapter: Text compatible image prompt adapter for text-to-image diffusion models. arXiv, 2023.
>
>
> ---
>
> **Q5: On the concatenation baseline and the role of $\alpha$.**
>
> **A5:**  We thank the reviewer for this suggestion. Following it, we implemented a weighted concatenation variant $z_{\mathrm{cat}}(\alpha) = \mathrm{concat}(\alpha z_1,(1-\alpha) z_2)$  and visualized its outputs for different $\alpha$ (Fig 16).
>
> As shown in Fig 16, this variant indeed requires much larger changes in $\alpha$ (e.g., from $0.1$ to $0.8$) to visibly affect the result, confirming that $\alpha$ has a weaker influence in the concatenation space than in the interpolated space. More importantly, for all tested $\alpha$ values the fusion behaviour remains *stitching-like*: one region of the image is dominated by $I_1$ (strawberry) and another by $I_2$ (jar), with a clear boundary between them. This suggests that keeping $z_1$ and $z_2$ in separate blocks encourages the network to treat them as two glued pieces rather than a single coherent object.
>
> By contrast, our MDeNoise stage mixes $z_1$ and $z_2$ via spherical interpolation within the *same* latent subspace. As shown in Fig 5 and our main results, this design produces much more integrated hybrids with smoothly shared geometry and appearance, which supports our choice of slerp-based mixing in MDeNoise over concatenation-based schemes.

---

> ### Author Response · Authors · 2025-11-21
>
> **Q6: Ablation to isolate BNoise and MDeNoise contributions.**
>
> **A6:** We appreciate the reviewer’s request for a clearer ablation of BNoise and EAA. Before detailing the new experiments, we first clarify what we mean by *semantic noise*. In our BNoise stage, we do not simply resample a different Gaussian; instead, we start from pure noise $\epsilon \sim \mathcal{N}(0,\mathbf{I})$, run a short *conditional* denoising process with the scale-concatenated image embeddings $\mathrm{SCat}(z_1,z_2;\beta_1,\beta_2)$ and the guiding prompt $P_G$, and then invert this process back to the noise level:  $\epsilon_b = \mathrm{Inv}(\mathrm{Denoise}(\epsilon \mid z_1,z_2,P_G)).$  Because this denoise–invert cycle is conditioned on both $I_1$ and $I_2$, the resulting $\epsilon_b$ lies closer to the joint data manifold of the two objects and carries semantic/structural cues from both sources, rather than behaving as an unconditional Gaussian sample.  Fig 4 shows that initializing from $\epsilon_b$ already yields fused structures that are more coherent than those obtained from random Gaussian noise, supporting this interpretation of BNoise as *semantic* noise.
>
> To quantify the effect of this semantic noise and of the EAA search, we conducted additional ablations on 1184 pairs sampled from IIOF (Fig 14, Table 4). The settings are:
> - **Baseline 1**: random noise $+\,$MDeNoise with fixed $\alpha=0.5$ (no BNoise, no search).
> - **Baseline 2**: Baseline 1 $+\,$BNoise with $\beta_1=\beta_2=1$ (semantic noise injected, no search).
> - **Baseline 1 + $\beta_1,\beta_2$-search**: Baseline 1$+ $EAA search over $(\beta_1,\beta_2)$, where different $(\beta_1,\beta_2)$ control how $I_1$ and $I_2$ contribute to BNoise.
> - **Random noise + $\alpha$-search**: Baseline 1$+ $EAA search over $\alpha$ (no BNoise).
>
> Qualitatively (Fig 14), **Baseline 1** often loses information from one of the sources, while **Baseline 2** preserves more structures from both inputs, indicating that the semantic noise $\epsilon_b$ provides a better initialization than pure Gaussian noise. Quantitatively, this is reflected by small but consistent gains of SA/SCE scores when going from Baseline 1 to Baseline 2 in Table 4.
>
> Enabling **$\beta$-search**(Baseline 1 + $\beta_1,\beta_2$-search) further improves all SA/SCE and SS scores and, importantly, reduces $B_{\text{sim}}$ from $0.691$ (Baseline 1) to $0.553$. Intuitively, EAA first uses $\alpha$-search to find a good balance between the mixed image embeddings under the *current* noise, and then uses $\beta$-search to adjust $(\beta_1,\beta_2)$, thereby rebalancing how each source contributes to BNoise and effectively enlarging the search space when the current balance is insufficient.
>
> By contrast, **Random noise + $\alpha$-search** can indeed find a better trade-off between the two image embeddings than Baseline 1 (higher SA/SCE and lower $B_{\text{sim}}$), but the lack of a semantically informed noise initialization means that fine details and parts from the inputs are more likely to be missing, as can be seen in Fig 14.
>
> Overall, these results clarify the roles of the two components: (i) BNoise produces a *conditional*, information-carrying noise $\epsilon_b$ via the denoise–invert cycle, rather than merely resampling Gaussian noise; and (ii) $\alpha$-search finds a good balance point in the mixed image embeddings during MDeNoise, while $\beta$-search rebalances the semantic contributions of $I_1$ and $I_2$ in the noise itself when necessary. Their combination yields more balanced and faithful hybrids than either component alone, which motivates our full HSP+EAA design.

---

> > ### Author Response · Authors · 2025-11-26
> >
> > Thank you very much for your careful review and valuable comments on our manuscript. We have revised the paper and provided a point-by-point response in accordance with your suggestions. We would like to kindly confirm whether our revisions have sufficiently addressed your concerns. If there are any concerns, please do not hesitate to let us know, and we will be glad to make additional improvements. We sincerely look forward to your feedback.

---

### Official Review · Reviewer_S3DV · 2025-10-31

**Soundness:** 3
**Presentation:** 3
**Contribution:** 3
**Rating:** 6
**Confidence:** 5

**Summary:**

The paper presents VMDiff, a inference-time optimization framework for two image visual mixing using FLUX-Krea. When attempting visual mixing with existing image generation models, they often struggle with coexistence and bias issues, resulting in conceptually disjoint images. To address these problems, the authors introduce Hybrid Sampling Process (HSP) and Efficient Adaptive Adjustment (EAA). To address these issues, the authors introduce Hybrid Sampling Process (HSP) and Efficient Adaptive Adjustment (EAA). The Hybrid Sampling Process (HSP) includes Blending Noise (BNoise), which concatenates two latents during the inversion process to preserve the original information of both images, and Mixing Denoise (MDeNoise), which interpolates the two latents to fuse the images. Efficient Adaptive Adjustment (EAA) searches the optimal $\alpha$ and then adjusts $\beta_1$, $\beta_2$ based on Similarity Score (SS) between the input images and texts, which are used as parameters in MDeNoise. Experiments demonstrate the effectiveness of the proposed framework for visual mixing on the newly proposed IIOF (Image-Image Object Fusion) dataset, both quantitatively and qualitatively.

**Strengths:**

1. The paper presents strong qualitative results demonstrating visually compelling outputs.
2. The training-free approach built upon FLUX-Krea is practical.
3. The experiments show that Blending Noise and Mixing Denoise are sufficient to achieve effective visual concept mixing.
4. A diverse set of experiments provides thorough analysis and validation of the proposed methods.

**Weaknesses:**

1. The citation format in paragraphs appears inconsistent with the conference’s official style guidelines, which significantly reduces readability.
2. I wonder whether the proposed method is specifically tailored to FLUX-Krea or if it can also be applied to other flow-based models, potentially with different optimal parameters. Demonstrating such generalizability would substantially strengthen the paper and highlight the broader applicability of the approach.
3. The proposed evaluation dataset covers a limited range of categories (40 selected classes), though this limitation is not critical to the overall contribution.
4. In Lines 79–80, the paper seems to reference the wrong figure example. The sentence “For instance, OmniGen Xiao et al. (2025) generates the doll figurine while entirely neglecting the horse.” likely refers instead to DreamO Mou et al. (2025).
5. In Line 373, there is a typographical error, an extra comma.

**Questions:**

1. How much GPU memory is required to mix one pair of images?

**Details Of Ethics Concerns:**

No concern.

---

> ### Author Response · Authors · 2025-11-21
>
> We thank reviewer S3DV for the clear summary and positive evaluation of our framework, qualitative results, and experimental design. In the revised PDF, we have corrected citation formatting and typos, and added new experiments/visualizations on backbone generalization and dataset coverage. Below, we address the comments on citation style, applicability beyond Flux-Krea, category range of IIOF, minor errors, and GPU memory usage.
>
> ---
>
> **Q1: On citation style, misreferenced example, and typo.**
>
>
> **A1:** We thank the reviewer for carefully checking these details. We have corrected all citation formats to strictly follow the official conference style. The sentence at Lines 79–80 now correctly refers to DreamO  instead of OmniGen, consistent with the figure example, and the extra comma at Line 373 has been removed. These fixes are reflected in the revised PDF.
>
> ---
>
> **Q2: Backbone generalization.**
>
> **A2:** This is an important point and we agree that the backbone should be decoupled from the proposed fusion mechanism. Conceptually, HSP+EAA is backbone-agnostic: it only assumes access to image-conditioned embeddings that can be mixed, and does not depend on any Flux-specific architecture. In practice, however, the $interface$ through which image information is injected is crucial for how much instance-level detail our fusion can preserve.
>
> Fig 12 applies our framework to three backbones: Flux-1.0-dev with the Redux image encoder, SDXL+IP-Adapter, and SD-3.5+IP-Adapter. With the same image pairs, fusion prompt, and hyperparameters, all backbones can often generate single hybrid objects, indicating that our method is not tailored to Flux-Krea alone. Flux+Redux encodes images into the same semantic latent space as text, so our BNoise+SInp mixing directly operates on rich, text-like image embeddings and best retains object-specific geometry and appearance from both sources. For SDXL and SD-3.5, image information is provided via IP-Adapter [ref-1], which keeps the text prompt fixed and injects image features as additional attention tokens; interpolating these tokens mainly modulates coarse, prompt-level semantics and leads to more loss of fine instance details.
>
> Overall, these results suggest that VMDiff is most effective on backbones that (i) preserve rich instance-level information from the inputs, and (ii) expose this information as compact image embeddings that live in a semantic space compatible with text (as Redux does). Our SDXL/SD-3.5 experiments should therefore be viewed as feasibility checks under a weaker image interface rather than as the ideal deployment setting. We expect that equipping SDXL, SD-3.5, or other flow models with a Redux-style semantic image encoder could substantially narrow the fusion-quality gap to Flux-Krea, and may enable performance comparable to our Flux-based results.
>
> [ref-1] Ye H, Zhang J, Liu S, et al. Ip-adapter: Text compatible image prompt adapter for text-to-image diffusion models. arXiv, 2023.
>
> ---
>
> **Q3: How much GPU memory is required to mix one pair of images?**
>
> **A3:** For our default setting (mixing a single pair of images to produce one fused output), the peak GPU memory consumption is about 42975 MB ($\approx 42$\,GB). After quantizing the Flux–Krea backbone, the peak usage drops to about 31729 MB ($\approx 31$\,GB).

---

> ### Comment · Reviewer_S3DV · 2025-11-25
>
> Thanks for the authors’ response. I have read it as well as the other reviews. The rebuttal has sufficiently addressed my concerns. I was already leaning positive about this work, and I will maintain my previous rating of 6.

---

> > ### Author Response · Authors · 2025-11-25
> >
> > Thank you very much for your thoughtful follow-up. We’re glad that the rebuttal has addressed your concerns, and we sincerely appreciate your decision to maintain the positive rating. Your feedback has been very helpful and encouraging for us.

---

### Official Review · Reviewer_EaME · 2025-11-01

**Soundness:** 3
**Presentation:** 3
**Contribution:** 3
**Rating:** 6
**Confidence:** 3

**Summary:**

This paper presents VMDiff, a diffusion-based framework for visual mixing: fusing two input images into a single, coherent hybrid object rather than mere juxtaposition or style transfer. The method has two pillars: (1) a Hybrid Sampling Process (HSP) that first builds BNoise (guided denoising + inversion over a scale-concatenated pair of visual embeddings) and then performs MDeNoise with spherical interpolation of the two image embeddings, and (2) an Efficient Adaptive Adjustment (EAA) that automatically searches fusion parameters
using a Similarity Score balancing visual and semantic similarity and balance terms. The authors also introduce IIOF, a new benchmark and report strong quantitative metrics, extensive ablations, and user studies showing clear preference for VMDiff over recent multi-concept and mixing/editing baselines.

**Strengths:**

1. The task is interesting. The paper targets object-level fusion—a regime where many image editing/compositional models struggle (coexistence or bias toward one concept).

2. Well-designed fusion pipeline. The BNoise + MDeNoise split is principled: concatenation before inversion preserves fine details; spherical interpolation during denoising yields seamless integration.

3. The authors conduct a thorough evaluation. A new benchmark (IIOF) + diverse baselines, VQA- and critic-based scores, balance metrics, extensive ablations, and user studies strengthen empirical claims.

4. The authors provide Insightful analysis. The paper’s BNoise vs. MDeNoise discussion and concatenate vs. interpolate design choices are well argued and supported by qualitative/quantitative evidence.

5. The presentation, figures, and algorithm descriptions are clean and easy to follow.

**Weaknesses:**

I do not have too many concerns. My main concerns are about the computational cost and robustness of the whole pipeline for in-the-wild input.

1. The multi-stage HSP + EAA (multiple generations per search step, occasional resampling) can increase inference latency and compute cost, limiting practical throughput.

2. Robustness and failure analysis are light. The paper lacks a systematic robustness study and only briefly touches on failure cases (e.g., large semantic/structural gaps).

**Questions:**

Will you release the model and the data? It would be helpful for the image editing community.

---

> ### Author Response · Authors · 2025-11-21
>
> We thank Reviewer EaME for the positive assessment of our problem formulation, fusion pipeline, and experimental evaluation. We have updated the main paper and appendix with additional analyses and figures addressing the raised concerns. Below, we respond to the comments on computational cost, robustness for in-the-wild inputs, and our release plan for the model and dataset.
>
> **Q1: On the runtime of HSP+EAA.**
>
> **A1:** We acknowledge that the multi-stage HSP+EAA performs an iterative search over a small parameter set $\theta=\{\alpha,\beta_1,\beta_2,\epsilon\}$, which increases cost compared to one-shot sampling. However, this overhead is acceptable for our offline creative-design setting and can be alleviated in practice:
>
> - **Early stopping and interactive use.**
>   EAA is not an unbounded search: once the score $S(\theta)$ exceeds an acceptance threshold $T_h$, we terminate the procedure. In an interactive tool, intermediate fusion candidates produced during HSP+EAA can be shown to the user, who may stop the process as soon as a satisfactory result is obtained.
>
> - **Orthogonality to sampler and backbone.**
>   HSP+EAA is agnostic to the underlying diffusion sampler. Faster Flux-based inference engines such as Sglang Diffusion [ref-1], which provide higher inference throughput, can be directly plugged in to reduce the effective denoising cost per candidate. Likewise, future schedulers or a lightweight learned predictor that approximates EAA in a single pass would be complementary to our formulation and can further reduce runtime without changing the core fusion objective.
>
> [ref-1] Zheng L, Yin L, Xie Z, et al. Sglang: Efficient execution of structured language model programs. Advances in Neural Information Processing Systems (NeurIPS 2024).
>
> ---
>
> **Q2: Robustness to semantic/structural gaps and failure modes.**
>
> **A2:** We agree that robustness and failure modes are important. To provide a more systematic view, we analyze all 1560 image pairs in IIOF by relating the final fusion score $S(\theta)$ to the similarity between the two input images. For each pair, we compute a DINO-based image similarity $\mathrm{sim}(I_1,I_2)$ and group the results into three bins according to $S(\theta)$:
>
> - low score $S<1.8$: mean similarity $\overline{\mathrm{sim}} = 0.176$;
> - medium score $1.8 \le S \le 2.4$: mean similarity $\overline{\mathrm{sim}} = 0.210$;
> - high score $S>2.4$: mean similarity $\overline{\mathrm{sim}} = 0.389$.
>
> This monotonic trend shows that $S(\theta)$ behaves sensibly with respect to the intrinsic difficulty of the pair: VMDiff tends to produce higher-quality, more balanced fusions when the two inputs share some visual/semantic relatedness, and the best results arise when the source images are more compatible (e.g., figurine–figurine pairs with $\mathrm{sim}>0.5$ are typically fused into a single hybrid). Conversely, pairs with very low image similarity correspond to genuinely challenging cases where large semantic/structural gaps make any object-level fusion difficult; in this regime, VMDiff can still produce hybrids but more failures (e.g., distorted shapes or imbalanced contributions) are observed, as illustrated in our failure examples in Fig 20.
>
> Importantly, this analysis suggests that $S(\theta)$ itself can serve as a reliability indicator in practice: low scores flag hard, low-similarity pairs where the fusion should be treated with caution, while medium/high scores correspond to robust, visually coherent hybrids.
>
> ---
>
> **Q3: Release plan for VMDiff code and the IIOF dataset.**
>
> **A3:** Yes. We intend to release the full implementation of VMDiff as well as the IIOF dataset upon acceptance of the paper, so that the community can reproduce our results and build upon our framework for future image editing and fusion research.

---

> > ### Author Response · Authors · 2025-11-26
> >
> > Thank you very much for your careful review and valuable comments on our manuscript. We have revised the paper and provided a point-by-point response in accordance with your suggestions. We would like to kindly confirm whether our revisions have sufficiently addressed your concerns. If there are any concerns, please do not hesitate to let us know, and we will be glad to make additional improvements. We sincerely look forward to your feedback.

---

> > > ### Comment · Reviewer_EaME · 2025-11-28
> > >
> > > Thanks for the response. The additional analysis of efficiency and robustness addressed my concerns. I tend to maintain my score.

---

> > > > ### Author Response · Authors · 2025-11-28
> > > >
> > > > Thank you very much for your thoughtful follow-up. We’re glad that the rebuttal has addressed your concerns, and we sincerely appreciate your decision to maintain the positive rating. Your feedback has been very helpful and encouraging for us.

---

### Author Response · Authors · 2025-12-03
**Reviewer–Author Discussion Summary**

Dear Area Chair,

We would like to **thank you and the reviewers** for the time and effort devoted to evaluating our submission. Below we briefly summarize, at a global level, how our rebuttal and revision address the reviewers’ comments.

At the **global level**, we structured the rebuttal as a point-by-point response to all four reviewers, explicitly linking each concern to concrete clarifications, new experiments, or textual revisions in the paper. We kept a consistent narrative of VMDiff as a training-free, plug-and-play fusion framework, and carefully described both its strengths and limitations. In particular, we:
- clarified the definition and intent of our semantic fusion metrics (SA/SCE/SS) and their relationship to visual realism;
- added a systematic robustness analysis over the full IIOF set, relating the final fusion score to image similarity;
- extended the empirical study to multiple backbones (Flux+Redux, SDXL+IP-Adapter, SD-3.5+IP-Adapter);
- provided a component-wise runtime and GPU-memory breakdown for HSP+EAA;
- ran additional ablations to disentangle the roles of BNoise, MDeNoise, and α/β-search, as well as preliminary three-image fusion trials;
- corrected citation formatting and minor errors.

At the **reviewer level**, we tailored our responses while keeping this overall structure:

- For **Reviewer EaME**, we focused on efficiency and robustness, adding a runtime breakdown, a GPU-memory analysis, and a robustness–difficulty study over all IIOF pairs. We clarified the offline/interactive usage scenario and the relation to faster Flux-style engines, and explained how our framework can directly benefit from faster backbones and samplers.

- For **Reviewer S3DV**, we addressed citation style, dataset coverage, backbone generalization, and GPU memory. We added new results on multiple backbones and explicitly discussed when our formulation is most advantageous, thereby refining the scope and applicability of the method.

- For **Reviewer wsRz**, we expanded the discussion of evaluation design, clarified what our metrics measure versus what human raters judge, detailed the shared prompt settings across methods, and added new ablations that isolate BNoise vs. MDeNoise vs. α/β-search. We also discussed training-free baselines, backbone dependence, commutativity (text order), and concatenation-based variants.

- For **Reviewer 4syL**, we added preliminary multi-image experiments, cross-model evaluation, a more detailed time-cost analysis, and a discussion of which kinds of manual / fine-grained control our current design can reliably provide, and which remain limitations and future work.

After the rebuttal, **three reviewers (EaME, S3DV, and 4syL)** posted follow-up comments indicating that **their major concerns have been addressed** and that they maintain a **positive recommendation** (with increased confidence in one case). The remaining reviewer (wsRz) has not yet posted a post-rebuttal comment, but we have responded point-by-point to all of their questions in the public rebuttal and incorporated the corresponding revisions into the manuscript.

We would appreciate it if our clarifications and additional results could be taken into consideration during your final recommendation.

Thank you again for your time.

**Best regards,**

Authors of Paper ID-6942

---

### Meta-Review · Area_Chair_Ydri · 2026-01-07

**Summary:**

This paper introduces VMDiff, an inference-time diffusion framework for visual mixing that fuses two input images into a single, coherent hybrid object, addressing common failure modes such as object coexistence and dominance bias in prior methods. The approach requires no retraining and consists of two key components: a Hybrid Sampling Process (HSP) that combines noise-level blending and latent-space mixing via spherical interpolation to integrate visual cues from both inputs, and an Efficient Adaptive Adjustment (EAA) module that automatically optimizes fusion parameters using a similarity score balancing visual fidelity, semantic consistency, and symmetry. The authors also present IIOF, a new benchmark for image–image object fusion. Experiments, ablations, and user studies demonstrate that VMDiff consistently outperforms recent multi-concept and visual mixing baselines in both quantitative metrics and human preference.

**Reviewer Concerns:**

Reviewers raised the following main concerns:
	1.	Efficiency and computational cost: The multi-stage HSP + EAA pipeline can incur high inference latency and compute cost, and the paper lacks a detailed breakdown or ablation of time, memory usage, and component-wise overhead. (Reviewers EaME and 4syL)
	2.	Robustness, failure analysis, and evaluation clarity: Robustness to in-the-wild inputs and failure cases is insufficiently studied, and some quantitative results appear inconsistent or ambiguous across tables and figures, reducing confidence in the evaluation. (Reviewers EaME and wsRz)
	3.	Generality and extensibility: It is unclear whether the method generalizes beyond FLUX-Krea to other diffusion backbones, or extends beyond pairwise image fusion to multi-image settings. Demonstrating broader applicability would strengthen the contribution. (Reviewers S3DV and 4syL)
	4.	Baseline selection and comparison scope: Comparisons with baselines targeting different problem scopes may be suboptimal, and the paper would benefit from comparisons with more directly relevant or training-free methods. (Reviewer wsRz)
	5.	Presentation and minor issues: There are citation formatting inconsistencies, minor typos, incorrect figure references, and a relatively limited category coverage in the evaluation dataset. (Reviewer S3DV)

Reviewer wsRz was the only reviewer who gave a negative score. However, their concerns mainly focus on requesting additional baseline analyses and questioning the marginal gains of one component of the framework. They also suggested comparing with more training-free approaches, but did not specify which additional methods to include.

**Reviewer Scores:**

This submission received scores of 6, 6, 4, and 6. The concerns raised by the only negative reviewer were largely addressed, and the reviewer would likely have raised their score if the post-rebuttal author–reviewer discussion had not been terminated early. The AC also agrees that the proposed training-free approach for visual mixing is worth being presented at ICLR and therefore recommends acceptance.

---

### Decision · Program_Chairs · 2026-01-26

Accept (Poster)